# Emergent regulation of ant foraging frequency through a computationally inexpensive forager movement rule

**Lior Baltiansky[†], Guy Frankel[†], Ofer Feinerman***

Department of Physics of Complex Systems, Weizmann Institute of Science, Rehovot, Israel

**Abstract** Ant colonies regulate foraging in response to their collective hunger, yet the mechanism behind this distributed regulation remains unclear. Previously, by imaging food flow within ant colonies we showed that the frequency of foraging events declines linearly with colony satiation (Greenwald et al., 2018). Our analysis implied that as a forager distributes food in the nest, two factors affect her decision to exit for another foraging trip: her current food load and its rate of change. Sensing these variables can be attributed to the forager's individual cognitive ability. Here, new analyses of the foragers' trajectories within the nest imply a different way to achieve the observed regulation. Instead of an explicit decision to exit, foragers merely tend toward the depth of the nest when their food load is high and toward the nest exit when it is low. Thus, the colony shapes the forager's trajectory by controlling her unloading rate, while she senses only her current food load. Using an agent-based model and mathematical analysis, we show that this simple mechanism robustly yields emergent regulation of foraging frequency. These findings demonstrate how the embedding of individuals in physical space can reduce their cognitive demands without compromising their computational role in the group.

*For correspondence:
ofer.feinerman@weizmann.ac.il

[†]These authors contributed
equally to this work

**Competing interest:** The authors declare that no competing interests exist.

## Editor's evaluation

This valuable study is of relevance to the field of collective animal behaviour. The proposed crop-cue-based motion-switching rules provide a welcome alternative to other models that assume far more deliberative abilities of ants. The authors present solid evidence to back up their claims.

## Introduction

Ant colonies rely on individual cognition and communication networks to perform complex collective tasks (*Feinerman and Korman, 2017*). Since brain tissue requires significant energetic investment (*Niven and Laughlin, 2008*), there is an advantage to communication systems that reduce the cognitive burden of the individual (*Lihoreau et al., 2012*). Stigmergy is a form of communication that can reduce individual cognitive demands by utilizing the physical environment (*Grasse, 1960*). It is the basis of some seminal examples of collective task performance in social insects, such as pheromone trail formation and nest construction (*Theraulaz and Bonabeau, 1999*; *Theraulaz et al., 1998*). In these examples, individuals alter the environment (i.e. lay a pheromone or dig a tunnel) in response to environmental changes made by other individuals, in such a way that leads to the emergence of the collective phenomenon. It has recently been suggested that the spatial properties of the physical environment, coupled with the individuals' form of movement in that environment, can also be utilized to offload computation from individuals' cognition to their environment. This form of communication has been described in the context of collective quorum sensing for the collective task of nest selection

(*Pavlic et al., 2021*). The same principle may apply to other systems in collective behavior, including foraging regulation.

Foraging in ant colonies is carried out by a small fraction of the workers, called foragers (*Oster and Wilson, 1978*; *McCook, 1880*). When the foragers return to the nest, they distribute their harvest to other ants in the nest, and then re-exit to collect more food (*Traniello, 1977*). This repetitive process persists as long as the food source is not exhausted or the colony satiates. Intriguingly, the rate at which food enters the colony by individual foragers matches the total level of hunger in the colony (*Buffin et al., 2009*; *Sendova-Franks et al., 2010*; *Greenwald et al., 2018*), implying the existence of a cross-scale feedback. The decentralized nature of the ant colony dictates that this regulation emerges from local actions of individual ants.

Liquid food, such as honeydew or nectar, is commonly carried within an ant's crop, an organ specialized for storing predigested food (*Eisner and Wilson, 1952*). From there it can be regurgitated to pass to other ants in mouth-to-mouth feeding interactions called *trophallaxis* (*Wilson and Eisner, 1957*). Trophallaxis is the main food-sharing method in many ant species (*Meurville and LeBoeuf, 2021*). It allows food to circulate through a complex trophallactic network among all colony members (*Sendova-Franks et al., 2010*; *Greenwald et al., 2019*; *Wikle et al., 2019*; *Quque et al., 2021*; *Bles et al., 2018*). In this paper, we focus on primary trophallactic interactions in which laden foragers returning to the nest, unload their crop contents to several receivers within the nest.

Previously, we used unique real-time measurements of fluorescent food inside the crops of all ants in a *Camponotus sanctus* colony, to infer quantitative links between local trophallaxis rules and the emergent regulation of food intake rate (*Greenwald et al., 2018*). The total level of hunger in the colony appeared to affect two aspects of the foragers' behavior. The first was the rate at which each forager unloaded her crop to receivers in the nest, which became slower as the colony satiated. Specifically, each forager's unloading rate was proportional to the total 'empty crop space' in the colony (hereinafter, 'colony hunger'). The second was the average frequency at which each forager exited the nest for foraging. These individual foraging frequencies were, on average, linear with 'colony hunger'. Our goal was to explore how these forager-colony relationships emerge from local rules.

The scaling of foragers' unloading rate to total colony hunger was quite comprehensively explained by local trophallaxis rules that were identified from the empirical data (*Greenwald et al., 2018*). Progress has also been made toward revealing the local rules that dictate a linear relation between average foraging frequency and colony hunger. However, the latter is understood to a lesser extent: No precise immediate cause for a forager to exit the nest has been found. Contrary to past assumptions (*Traniello, 1977*; *Gregson et al., 2003*; *Buffin et al., 2009*), foragers did not exit the nest only after they unloaded their entire crop contents, nor had we observed a clear crop-load threshold below which the foragers were more likely to exit. Rather, the foragers exited the nest with highly variable crop loads. Some studies have successfully identified local social triggers for the exits of foragers in several ant species (*Pinter-Wollman et al., 2013*; *Mailleux et al., 2011*; *Greene and Gordon, 2007*; *de Biseau and Pasteels, 2000*; *Davidson et al., 2016*; *Pless et al., 2015*; *Razin et al., 2013*; *Robinson et al., 2012*), but most have focused on foraging initiation, and not on the subsequent decay of activity in response to gradual colony satiation (*Rivera et al., 2016*). Our crop-load measurements in *Greenwald et al., 2018* have shed light on the local determinants of foragers' exits that linearly relate them to the current level of colony hunger. The local factors found to affect the temporal probability of a forager to exit the nest were both her instantaneous crop load and her unloading rate in the nest: the emptier her crop and the faster her unloading, the more likely a forager was to exit. A simple Markovian decision-making model that yields the observed results was proposed. However, with no empirical access to the exact timings at which the forager assesses her next decision of whether to stay in the nest or leave to forage, the assumptions of our model could not be verified.

Here, we present an alternative mechanism that reduces the cognitive demand on individual foragers through utilization of physical space. This mechanism is supported by previously unexplored aspects of the data produced by our past experiments. The individual crop load dataset is now enriched with detailed spatial tracking of the foragers inside the nest. Together, these point to a new behavioral rule. A clear transition between two movement modes, depending on the forager's instantaneous crop load, is evident from the new data: As foragers move around the nest, unloading their crops to ants that they meet, they tend to step toward the depth of the nest when their crop load is above a certain threshold, and tend to step toward the exit when it is below this threshold.

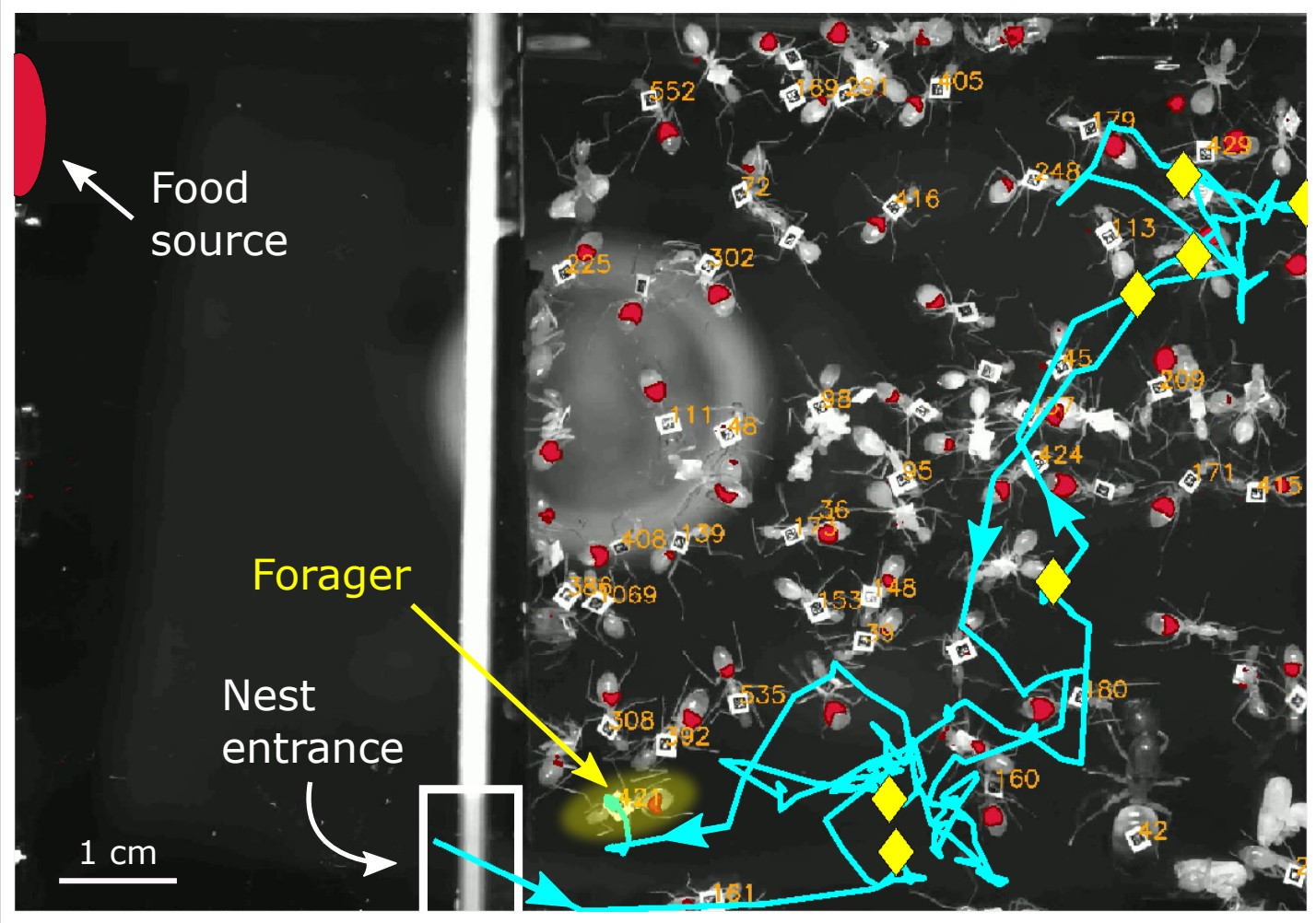

**Figure 1.** Example of a forager's trajectory in the experimental nest. A single frame from an experimental video shows the nest on the right and a foraging arena on the left, where a fluorescent food source was presented. The food source is marked as a red oval and the nest entrance is marked by a white rectangle at the bottom left corner of the nest. Ant IDs are presented next to their tags, and the imaged food in their crops is overlaid in red. A single forager is highlighted in yellow, and her trajectory from when she last entered the nest is presented in cyan. Arrows on the trajectory mark the directionality of her path, and yellow diamonds mark locations of trophallactic interactions that she performed in her unloading bout.

Since the movement on both sides of the threshold is highly stochastic, this transition is masked when looking at the ultimate exit probabilities, which was the approach taken in *Greenwald et al., 2018*. Here, we present an agent-based model that implements this new stochastic motion rule along with the previously reported trophallaxis rules, and analyze it mathematically. The results show that these rules suffice to shape a forager's trajectory in the nest in a way that produces linear foraging frequency regulation while maintaining low levels of individual cognitive loads.

## Results

### Foragers move according to a biased random walk that is crop state dependent

Starved colonies of *Camponotus sanctus* ants were recorded in an artificial 2D nest as they gradually replenished on fluorescently-labeled food. All ants were tracked, the amount of food in their crop was quantified throughout time using fluorescence imaging, and all trophallaxis events were annotated (*Greenwald et al., 2018*; *Baltiansky et al., 2021*). Foragers were identified to be those ants that repeatedly left the nest to retrieve food and deliver it to other ants in the nest. We analyzed the trajectories of these foragers inside the nest in relation to their changing crop state, as they distributed their

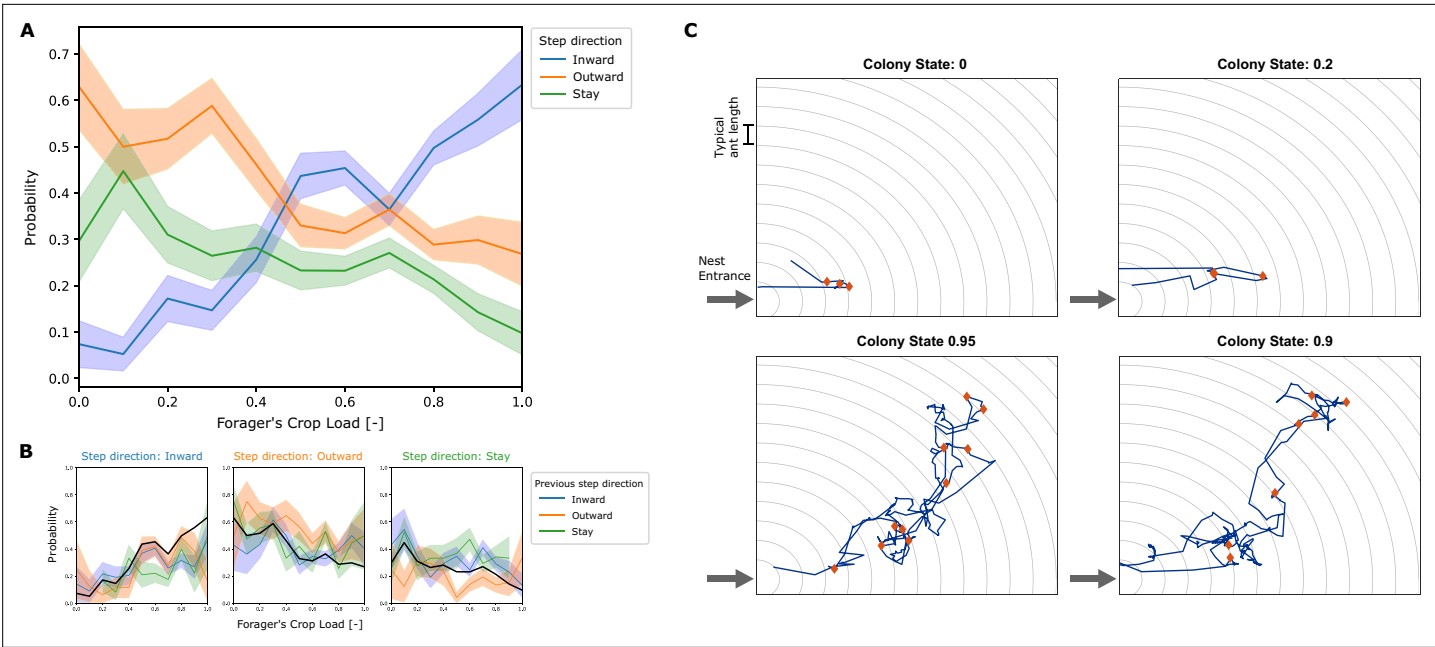

**Figure 2.** Empirical movement of foragers in the nest. (**A**) All locations in the nest were binned according to distance from the entrance, with bin width of 1 typical ant length (as visualized by circular grid lines in panel C). At each interaction of a forager, her crop load at the end of the interaction and the location bin of her next interaction was recorded. Pooled data from all foragers was used to calculate the probability of the next interaction to be in a deeper location bin (inward), in the same location bin (stay), or in a location bin closer to the entrance (outward), as a function of their crop load. Probabilities and standard deviations were calculated for each one of 10 crop load bins. Standard deviation was calculated by the formula for multinomial STD: $\sqrt{p(1-p)/n}$, and is represented by the error bars in the plot. Sample sizes for each one of the 10 crop load bins ($n$) are {27, 38, 58, 68, 78, 103, 185, 192, 187, 77, 41}. (**B**) The data described in panel A was grouped according to the direction of the previous step. The plots show the probability to step inward (left), outward (middle) and stay (right), for cases where the previous step was inward, outward or stay as different curves. The pooled probability for all previous directions is presented as a thick black curve, equivalent to the curves presented in panel A. Standard deviations were calculated as in panel A, sample sizes for each crop load bin ($n$) for each previous step direction are "inward": {11, 23, 26, 34, 46, 81, 73, 79, 26, 15}, "outward": {8, 16, 17, 17, 23, 43, 36, 37, 13, 6}, "stay": {19, 18, 24, 24, 19, 36, 51, 35, 9, 4}. (**C**) Examples of trajectories of single unloading bouts of a forager in the nest. Nest entrance is at the bottom left corner. Grid-lines spaced by a typical ant length are presented in gray. These are the spatial bins used to define the distance from the entrance for calculating the foragers' biases (panels A-B). The trajectory of the unloading bout is plotted in blue, and locations of trophallaxis events are presented as red diamonds. The top two plots present trajectories from low colony states, and the bottom two plots present trajectories from high colony states.

The online version of this article includes the following video and source data for figure 2:

**Source data 1.** Empirical data.

**Figure 2—video 1.** Forager 421's 12th unloading bout, when the colony was 90% full.

https://elifesciences.org/articles/77659/figures#fig2video1

**Figure 2—video 2.** Forager 421's 4th unloading bout, when the colony was 20% full.

https://elifesciences.org/articles/77659/figures#fig2video2

food in trophallactic interactions. *Figure 1* shows a single frame from an experimental video overlaid with an example of a forager's trajectory in the nest.

We found that the movement of a forager in the nest can be characterized by a random walk with a bias that depends on the amount of food in her crop (*Figure 2A*). At each trophallactic interaction that a forager performed, her distance from the nest entrance was measured. The probabilities of her next interaction to be farther from the entrance (step inward), closer to the entrance (step outward) or at the same distance from the entrance (stay), were calculated as a function of the forager's crop state at the end of the interaction. This coarse-grained analysis revealed a crop load threshold of 0.45 that separates between two types of movement. When the forager's crop load is higher than the threshold value, she is more inclined to step inward into the nest. Conversely, at lower crop loads she is more probable to step outward toward the exit (*Figure 2A*). *Figure 2B* shows that these probabilities are not affected by the direction of the forager's previous step. Thus, it is reasonable to model the forager's movement as a Markovian process.

Foragers that operate according to the crop-dependent movement described above are expected to generate random closed paths in the nest as they unload their crops via trophallaxis: since the forager steps into the nest with a relatively full crop after she fed at the food source, her initial bias drives her deeper into the nest. As she unloads her food to other ants, her crop load may reach a level at which her bias switches direction. The forager then continues to disseminate food to ants in the nest, but now with a drift that carries her toward the exit, until she finally reaches it and leaves the nest to forage again. Note that as the colony gradually satiates, the forager's unloading rate decreases (*Greenwald et al., 2018*). Therefore the duration and the depth of the forager's cyclic paths both rise with the colony's level of satiety. (Note that the experimental nest is flat, and the term 'depth' merely refers to distance from the entrance and not vertical depth.) *Figure 2C* shows examples of empirical paths of unloading bouts of individual foragers in the nest. When the colony is hungry (colony state close to 0), the paths are short and include few trophallactic interactions, and when the colony approaches satiation (colony state close to 1), paths are long and include more trophallactic interactions. *Figure 2—video 1* and *Figure 2—video 2* are fragments from an experimental video that show unloading bouts overlaid with the trajectory of the forager as she unloads in the nest at a high colony state and at a low colony state.

To explore whether this empirically-derived movement rule may underlie the fact that foraging frequency scales linearly with total colony hunger, we simulated it numerically and analysed it mathematically. In the next sections we present two agent-based models and an analytical description of the system. The first agent-based model mimics the experimental two-dimensional nest. The second model is a simplified 1-dimensional version which is more readily approachable analytically. Our simulations show that both models yield the desired linear foraging frequency regulation. We then solve the 1D model analytically to show how it accounts for this emergence based on the local movement rule described above. Finally, we compare different properties of these models to our empirical observations.

## Agent-based model in a 2D nest

This model implements a square 2D nest of size 11x11 ant-lengths which contains 89 nest ants. The size of the nest and the number of ants were chosen to be of similar scale to the experimental conditions. The nest has a single entry/exit, located at one of the corner cells, mimicking the structure of the experimental nest. A single forager loads her crop outside the nest and then enters the nest, moving around and distributing her food load to the nest-ants. For simplicity, the nest-ants only receive food from the forager and do not redistribute it further. The forager's movement is based on the empirical turning angles of foragers, such that is captures the two movement types described in *Figure 2*: generating an inward drift when her crop load is above the empirically identified threshold and an outward drift below it (see description below). When the forager happens upon the nest entrance, she exits the nest, refills her crop, and re-enters to distribute her new load. Note that contrary to the assumptions used in our previous paper (*Greenwald et al., 2018*), here a forager never directly decides to exit the nest. Rather, the forager only decides on the direction of her next step, and an exit occurs if the forager's motion brings her to the nest exit. Hereafter, we refer to all the steps between the forager's entrance and exit as a single *unloading bout*. For more details please refer to the Materials and Methods section.

The simulation implements three simple rules that were derived from the experimental data.

1. Forager movement. At each step of the simulation, the forager moves a distance of 0.2 ant-lengths (this step size is the average distance that foragers moved per second empirically). Upon entering the nest, and after each trophallactic interaction with a nest-ant, the forager randomly samples a new direction of movement from the empirical distribution of directions that actual foragers were observed to take. Two angle distributions were extracted from the experimental data (*Appendix 1—figure 1*): one distribution of directions taken by foragers when their crop load exceeded the threshold 0.45 (these directions tended to point inward away from the entrance), and the other of directions taken by the foragers when their crop load was below that threshold (these directions tended to point toward the exit). In the simulation, the forager sampled her new direction from the respective empirical distribution given her current crop load.

2. Trophallaxis. At every step, if there is a nest-ant within 0.2 ant-lengths (the ant's antennae reach) from the forager, the two perform trophallaxis. The amount of food passed from the forager

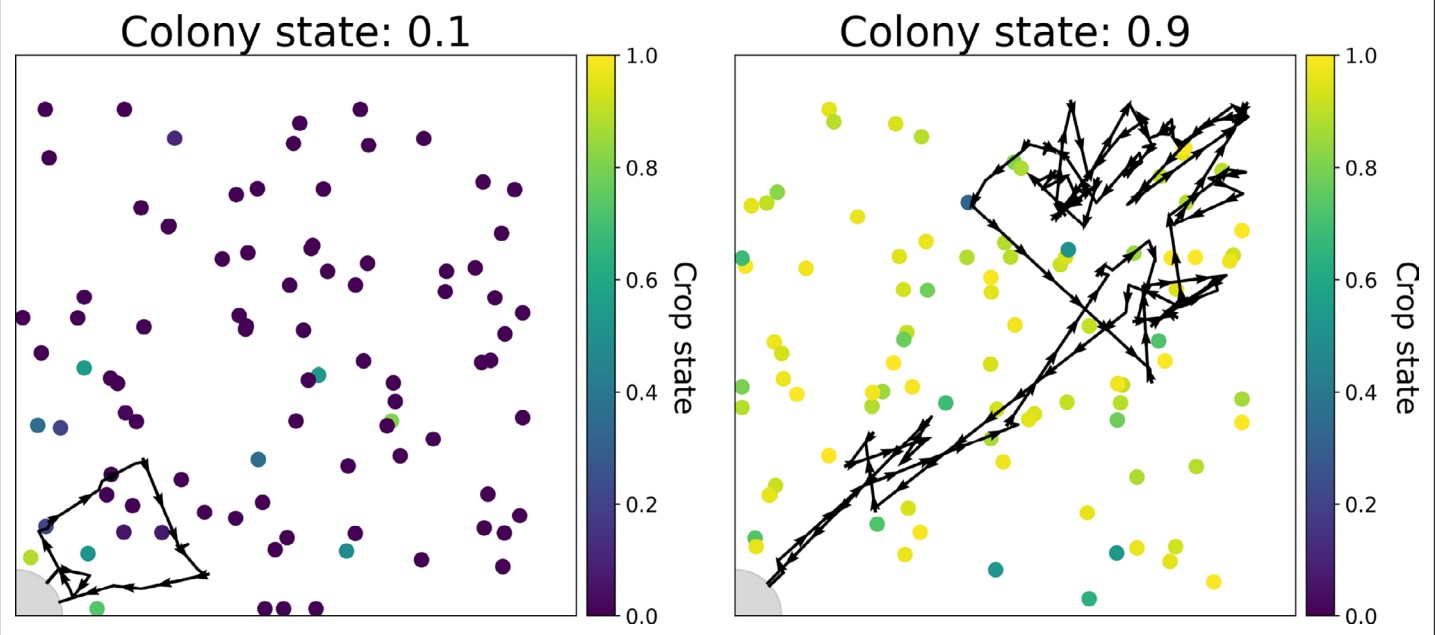

**Figure 3.** Examples of two unloading bouts of a forager in the 2D simulation. The 89 nest-ants are depicted by colored circles at their positions at the beginning of the bout. The color represents the ant's crop state (purple represents an empty crop, yellow represents a full crop). The nest entrance is marked by a gray area at the bottom left corner of the nest. All the forager's positions during the unloading bout are presented as a black trajectory through the nest, with arrows representing the forager's direction.

to the nest-ant is stochastic, and is scaled to the available crop space of the receiver ant. It is a random, exponentially distributed, fraction of the receiver's unfilled crop space, with an average of 0.15, as was observed empirically in *Greenwald et al., 2018*. If the forager has insufficient food, she gives all that she has.

3. Nest-ant movement. Nest-ant movement is implemented in the model as a random walk (at every step, each ant moves a distance of 0.2 ant-lengths in a random direction). Nest-ant movement contributes to the spatial homogenization of the food in the colony, which causes the forager to interact with ants that are, on average, representative of the satiety state of the whole colony (see section 3.5). This unbiased sampling was observed empirically, and together with the empirical trophallaxis rule, causes the forager to unload her crop at a rate proportional to the colony's total hunger level (*Greenwald et al., 2018*).

The simulation was run 200 times and qualitatively reproduced the lengthening and deepening of foragers' unloading bouts with colony satiation that were observed empirically (compare *Figure 3* and *Figure 2C*). *Figure 3* depicts two unloading bouts of the forager within the simulated nest, from different stages of a single run of the simulation: one from an early stage of the run, when the colony was 10% satiated, and the second from a later stage, when the colony was 90% satiated. These representative examples demonstrate how the same set of unloading and movement rules by which the forager operates, produces short trajectories when the colony is relatively hungry, and longer trajectories as the colony satiates.

Following the empirical analysis in *Greenwald et al., 2018*, 'foraging frequency' was calculated as the inverse of the duration of the forager's unloading bout in the nest. This shows that the dynamics of the lengthening of the trajectories in the nest indeed lead to a linear relationship between the average frequency at which the forager encounters the nest exit and the amount of food accumulated in the colony. This is analogous to the linear matching of foraging frequency to colony hunger that was observed empirically (*Figure 4A and B*).

Note that the model has reproduced the linear scaling between foraging frequency and empty colony state, but it was not expected to capture the exact values of the empirical observation. Quantitative discrepancies are a result of factors that were not incorporated into the model to avoid overcomplication, such as: nest-ant behavior (spatial distribution, movement and secondary trophallaxis

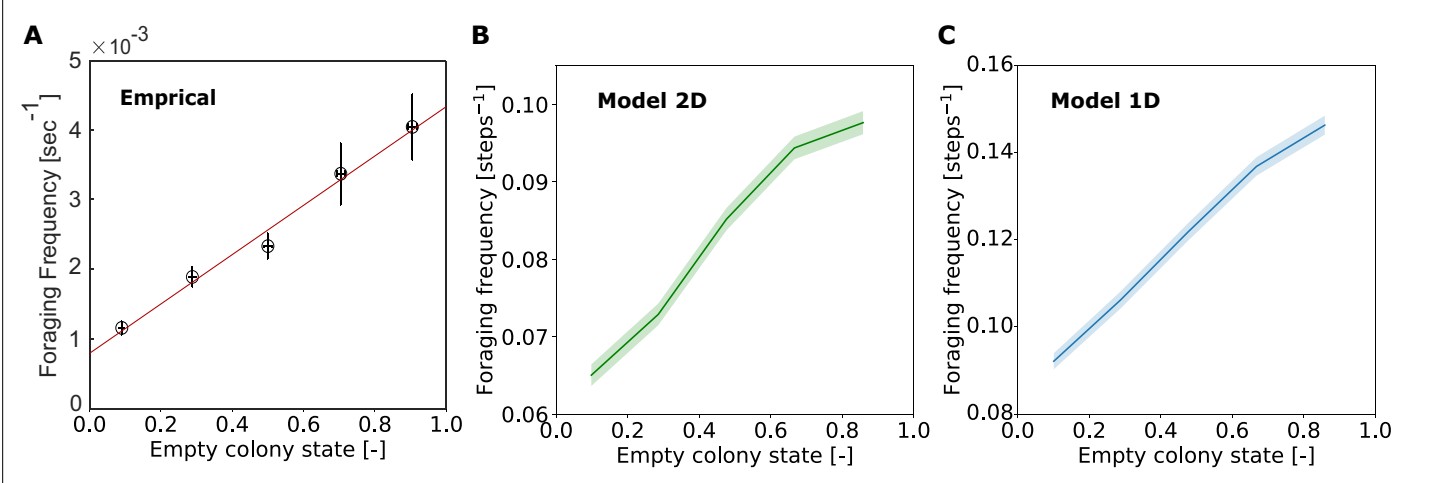

**Figure 4.** Foraging frequency scales linearly with empty colony state. Foraging frequency was calculated as the inverse of the duration of the forager's unloading bout in the nest. Unloading bouts were binned into five equally spaced bins of colony state, and the mean and SEM of foraging frequency was calculated for each bin. (**A**) Experimental data, figure taken from figure 4B of *Greenwald et al., 2018*. Data was grouped into equally-spaced bins of colony state (n = 57, 39, 28, 26, 26, for bins 1–5, respectively). (**B**) Data from 200 repeats of the 2D model simulation. Data from all repeats was pooled and grouped into equally-spaced bins of colony state (n = 3869, 4183, 4489, 4895, 6248, for bins 1-5, respectively). (**C**) Data from 200 repeats of the 1D model simulation. Data from all repeats was pooled and grouped into equally-spaced bins of colony state (n = 1770, 1989, 2222, 2531, 3189, for bins 1-5, respectively).

The online version of this article includes the following source data for figure 4:

**Source data 1.** Data from 1D model.

**Source data 2.** Data from 2D model.

between nest-ants), the duration of trophallaxis events, and the fact that there is more than one forager.

Hence, the three local rules of the agent-based model lead to the emergence of a colony-level regulation that is qualitatively consistent with the empirical foraging frequency regulation. Since the direction of the forager's movement in the nest is coupled to the amount of food that she carries, and given that her rate of unloading is determined by the amount of food in the receivers' crops, there emerges a negative feedback between the amount of food stored in the colony and the frequency at which foragers exit the nest to bring in more food. This cross-scale feedback, from the level of colony hunger to the level of individual foraging events, emerges with no need for the forager to sense anything but her own current crop load.

To understand how this linear scaling emerges from the local rules described above, we introduce an analytically tractable one-dimensional model. Since the direction of the forager's movement is defined relative to the nest entrance (toward the entrance, away from the entrance), the forager's position may essentially be defined using a single coordinate – her distance from the entrance. This description is further strengthened by the fact that coarse-graining the forager's motion in a single dimension reveals a clear threshold-dependency of her motion bias on her crop load (*Figure 2*). The nest can then be simplified to a one-dimensional array of nest-ants through which the forager walks back and forth. In the next section, we describe the 1D simplification of the agent-based model.

## Agent-based model in a 1D nest

This model implements a 1D nest consisting of 45 cells, each cell inhabiting one nest-ant. The

**Table 1.** Movement biases for agent-based model.

The probabilities of a simulated forager to step inward, outward or to stay in the same cell, for two cases: when her crop load is lower than or higher than a threshold (0.45). The values of the threshold and the biases are approximated based on the empirical data (*Figure 2A*).

| Crop load | P(inward) | P(outward) | P(stay) |
|---|---|---|---|
| ≤ 0.45 | 0.16 | 0.53 | 0.31 |
| > 0.45 | 0.46 | 0.32 | 0.22 |

point of entrance/exit of the nest is from one of its edges. A single forager walks in the nest and feeds nest-ants as described in the 2D model above, with the following adjustments:

1. Forager movement. At each step, the forager randomly chooses a direction of movement 'inward' - moves one cell away from the entrance, 'outward' - moves one cell toward the entrance, or 'stay' - stays on the same cell with probabilities that depend on her current crop load. Based on the empirical data (*Figure 2A*), the probabilities were set as defined in *Table 1*.
2. Trophallaxis. At every step, the forager performs trophallaxis with the nest-ant in her cell. The amount of food passed from the forager to the nest-ant is the same as described in the 2D model.
3. Nest-ant movement. All nest-ant positions are randomly shuffled between forager unloading bouts. This shuffling replaces a simple random walk in the 2D model. In 1D, this shuffling is required for sufficient homogenization to yield a representative sample of receivers for the forager, and is supported by a one-dimensional projection of the empirical nest-ant data (see details in Appendix 1).

For more details on the implementation of the 1D model, see Materials and methods.

*Figure 4C* shows that, similar to the 2D model, the 1D model reproduces the empirical observation that foraging frequency scales linearly with colony hunger. Next, we present a mathematical description of the 1D system to analytically explain these results.

## The emergence of linear scaling between foraging frequency and total colony hunger

A precise analytical description of the unloading bouts of a forager is challenging, since they involve stochasticity in her movement, in the amount of food she delivers at each interaction, and in the state of her nest-ant partners. Therefore, we use a coarse-grained analysis, where we consider the averages of these stochastic variables: the forager's average direction, the average amount of food given per interaction, and the average state of the forager's partners in an unloading bout.

Here, for the sake of simplicity, we present the equations for the deterministic case in which a forager walks only inward when her crop load exceeds the threshold, and only outward when it is below the threshold. In Appendix 1, we show how these equations apply to the more general case, where the forager's bias is set by the partial probabilities to walk in each direction.

Let $c$ denote the crop state of the forager ($c = 1$ when the forager's crop is full and $c = 0$ when it is empty). A certain crop load $c^*$ is the threshold that separates between the forager's two movement biases within the nest: the forager walks inward when $c > c^*$ and outward when $c < c^*$.

Let $F$ be the total satiety state of the colony ($F = 0$ when the colony is starved and $F = 1$ when all ants in the colony are satiated). The nest-ant movement rule dictates that the forager interacts with a representative sample of the colony at each unloading bout (*Greenwald et al., 2018*), such that the average state of the forager's partners is equal to the colony state, $F$. The trophallaxis rule gives the average amount of food delivered at each interaction: a fraction $\alpha$ of the receivers' empty crop space (*Greenwald et al., 2018*). Given these two rules, the average amount of food a forager unloads at every interaction is:

$$\langle \Delta c \rangle = \alpha \cdot (1 - F). \tag{1}$$

Since the forager enters the nest full, and since in the extreme case the forager performs trophallaxis with a new ant at each step, the average number of interactions she will make until her crop load reaches the threshold is $n^* = \frac{1-c^*}{\langle \Delta c \rangle}$. In the extreme case, this quantity is equivalent to the average position in the nest at which the forager switches her bias, denoted $\langle x_{switch} \rangle$. Therefore, we obtain the following relation between the colony state and average position at which the forager switches her bias:

$$\langle x_{switch} \rangle = \frac{1-c^*}{\alpha(1-F)} \tag{2}$$

The average duration of the foragers' unloading bout, $\langle T \rangle$ is the time it takes her to reach $\langle x_{switch} \rangle$ from the entrance (at $x = 0$) and return. In the extreme case, walking inward every step until $\langle x_{switch} \rangle$ and outward every step from $\langle x_{switch} \rangle$, this simply equals $2 \cdot \langle x_{switch} \rangle$. We get:

$$\langle T \rangle = \frac{2(1-c^*)}{\alpha(1-F)} \tag{3}$$

The frequency of the forager's unloading bouts, $R$, is defined as the reciprocal of the average unloading bout duration $\frac{1}{\langle T \rangle}$. Therefore, the foraging frequency is:

$$R = \frac{\alpha}{2(1-c^*)} \cdot (1 - F) = \mu \cdot (1 - F) \tag{4}$$

where $\mu = \frac{\alpha}{2(1-c^*)}$ is a constant. Thus, it is clear that the foraging frequency $R$ is proportional to the colony state of hunger $(1 - F)$. This is the linear relationship observed both experimentally and in the simulations of our agent-based models (*Figure 4*).

Clearly, in reality forager ants don't move in such an extreme manner within the nest, but the general logic of the analytical development above applies to a softer movement rule as well, where the forager's walk is more probabilistic. In short, the difference between an extreme walk and a probabilistic walk, means that the forager, going stochastically back and forth between the nest ants, may interact multiple times with the same ants before switching her bias. This distinction alters the average amount of food delivered at each step ($\Delta c$, *Equation 1*), as less food is given to a nest-ant with each subsequent encounter between her and the forager. Additionally, the number of interactions that it takes the forager to reach her threshold no longer translates directly to the position at which she switches her bias ($x_{switch}$, *Equation 2*). Nevertheless, the derivation in Appendix 1 shows that the average amount of food given to each nest-ant is still proportional to $(1 - F)$, and that since both the inward and outward biases are constant, the number of steps spent with each nest ant is, on average, also constant (neglecting boundary effects). Therefore, overall, the differences introduced by the probabilistic walk are expressed in the factor μ that multiplies the colony state of hunger $(1 - F)$ in *equation 4*. In the probabilistic movement case, μ is dependent on the fraction $\alpha$, the threshold $c^*$, and the probabilistic walking biases. For details, see Appendix 1.

## Further characteristics of unloading bouts in experiment and simulation

*Figure 5* presents additional dynamics that appeared in both the experimental and simulated data. The states of the foragers' recipients represent, on average, the states of all ants in the colony (*Figure 5A*). The forager unloads her food at a rate proportional to the empty colony state (*Figure 5B*). Furthermore, the forager's unloading bouts in the nest become deeper with colony satiation (*Figure 5C*), and the amounts of food in the forager's crop upon exiting the nest are highly variable at all colony states (*Figure 5D*). On average, they are relatively constant initially, and slightly rise at higher colony states.

While the empirical trends are captured by the models, the exact quantitative values do not necessarily match, since the models were not designed to capture the complexity of the whole colony-feeding system, as mentioned above. Additionally, there is a qualitative difference between the shapes of the empirical and simulated curves of the increasing depths (*Figure 5c*). While the empirical rise in depth is concave and seems to reach a plateau, the resulting curves of the agent-based models are convex. We speculate that three features of the empirical system that are not incorporated into the agent-based models may be the cause for this minor inconsistency. The first is that in reality ants occupy space in the nest, thus restricting the movement of other ants by steric interactions. That is, the forager's state space may be constrained since she may be blocked from reaching certain areas of the nest by other ants. On the other hand, in the model, the forager is able to walk over nest ants and hence has no state space restriction. The second feature is that the model does not implement trophallaxis between nest ants, whereas it is known that in real ant colonies nest ants do indeed spread food between themselves. Empirically, this nest ant behavior may be a reason that the forager does not need to cover all areas of the nest and may be a reason for the plateau in *Figure 5* Empirical C. Lastly, the third feature is the spatial distribution of ants in the nest. While in the 2D model nest ants are initiated in random positions and move around randomly, and in the 1D model nest ants occupy all cells in the nest, the empirical distribution of ants is usually characterized by dense regions of less mobile ants and sparse regions where ants tend to move more. Naturally, foragers' interactions with nest ants may occur only where nest ants are present, thus affecting the locations where foragers are found in the nest.

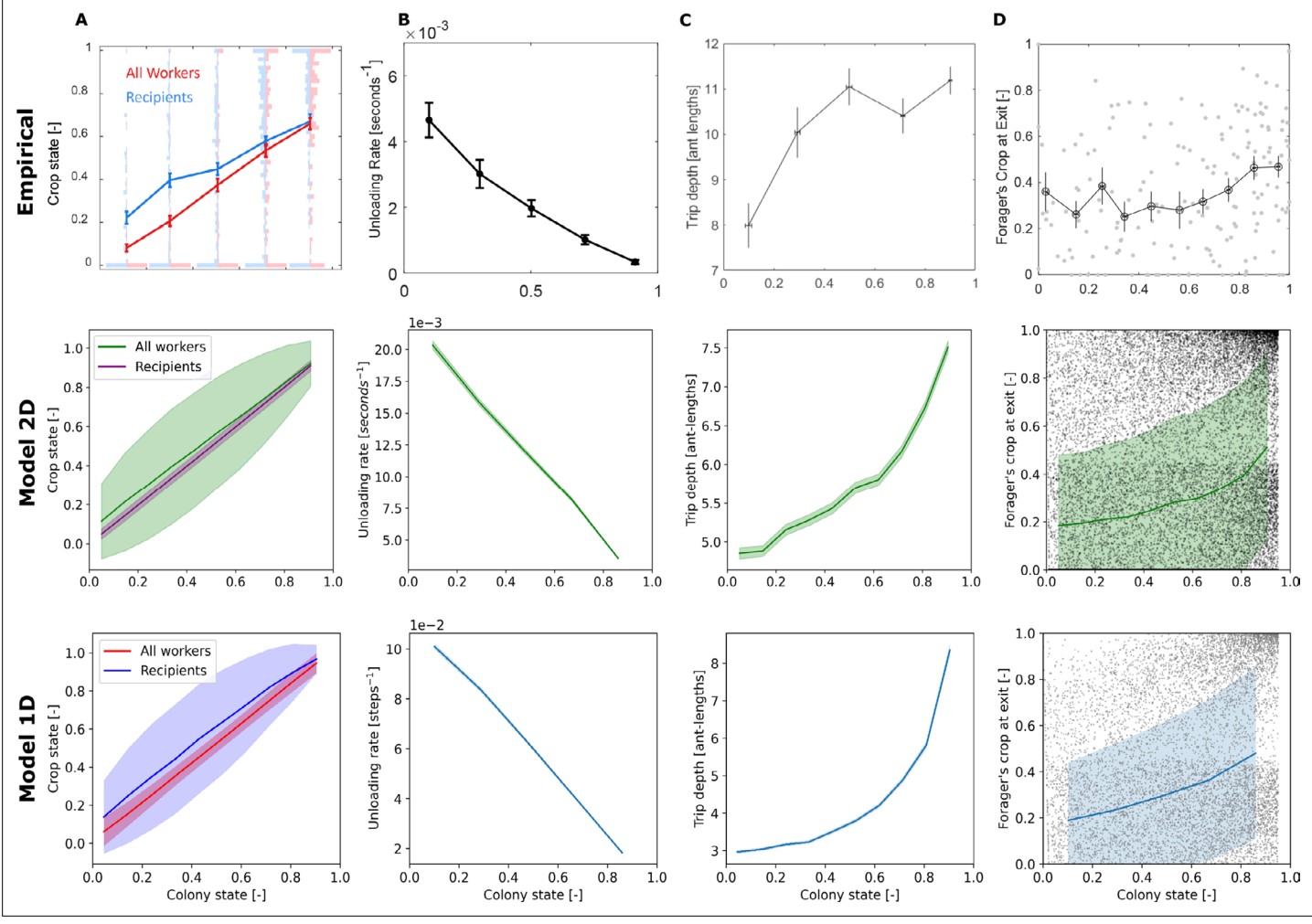

**Figure 5.** Comparison between empirical and simulation data of forager unloading bout dynamics. Unloading bouts were binned into equally-spaced bins of colony state. Means and SEMs of different measures were calculated for each bin. Plots of the empirical data were taken from Figures 3D and 2B (model predictions were removed), 5D (units were converted from mm to ant-lengths for comparability to simulation data), and 5A of *Greenwald et al., 2018*, respectively. Simulation data is from 200 replicates of each model. Sample sizes for each colony state bin are as specified in the caption of *Figure 4*. (**A**) The crop states of the nest-ants that interacted with the forager compared to the crop states of all nest-ants, averaged per unloading bout. Error bars for the simulated data represent STDs to better appreciate the variance of the distributions. (**B**) The forager's unloading rate, calculated as the amount of food she delivered in the unloading bout divided by the duration of the unloading bouts. (**C**) Depth, the maximal distance of the forager from the nest entrance in the unloading bout. (**D**) Forager's crop load at the end of the unloading bout. Error bars for the simulated data represent STDs to better appreciate the variance of the distributions.

## Discussion

Ant colonies manage to regulate foraging activity in response to their collective hunger, despite the fact that the foragers are only a small subset of the workers. In *Greenwald et al., 2018*, we have shown that as starved colonies gradually satiate, the average individual foraging frequency linearly matches the temporal state of hunger of the whole colony. Here, we have presented new experimental data that accounts for this relation between the colony scale and the individual scale. Combining spatial tracking of foragers within the nest with the dynamic measurements of their crop loads, our data imply a simple rule for the movement of foragers. These rules tie the forager's instantaneous crop load to the direction that they take within the nest. Overall, our findings suggest that a forager's trajectory in the nest can be shaped by the rate at which she unloads to recipient ants while this unloading rate is governed by the satiety of the recipients. Using an agent based model and mathematical analysis, we demonstrated that together with the trophallaxis rules described in *Greenwald et al., 2018*, this simple movement rule produces linear foraging frequencies as observed empirically.

Previous studies have identified local factors that may determine foraging activity in various species of social insects. These factors include chemical cues and the rate of interactions with other individuals (*Davidson et al., 2016*; *Pinter-Wollman et al., 2013*; *Mailleux et al., 2011*; *Greene and Gordon, 2007*; *de Biseau and Pasteels, 2000*; *Pless et al., 2015*; *Prabhakar et al., 2012*; *Pagliara et al., 2018*), the foragers' own nutritional state (*Toth et al., 2005*; *Mayack and Naug, 2013*), and larval hunger signals in the nest (*Howard and Tschinkel, 1980*; *Cassill and Tschinkel, 1995*; *Lee Cassill and Tschinkel, 1999*; *Cornelius and Grace, 1997*; *Pankiw, 2004*; *Dussutour and Simpson, 2009*; *Ulrich et al., 2016*; *Schultner et al., 2017*; *Ma et al., 2018*; *Kraus et al., 2019*; *Chandra and Kronauer, 2021*). Such factors were shown to relate the foragers' activity to external variables such as food quality or availability, and to the internal colony nutritional requirements (*Seeley, 1989*; *Cassill, 2003*). However, no study that we know of provides a full mechanistic explanation for the qualitative linear relationship between colony hunger and individual foraging frequency, which was observed during the process of gradual colony satiation. This striking linearity was revealed only recently, thanks to technological advances (*Greenwald et al., 2018*).

In our previous work (*Greenwald et al., 2018*), we have presented a model which described the forager's decision to exit as a function of both her crop load and her unloading rate; however, it did not present a comprehensive mechanism of action. In the current study, we show that the exiting rate dynamics can be described without an explicit decision to exit by the forager, rather, the forager only decides on the direction of her next step, leaving the nest to forage whenever she reaches the nest exit. Indeed, empirical data supports that such decisions are Markovian and depend on a single variable - the forager's current crop load.

One could compare the new model with the previous one in terms of the cognitive load required of an unloading forager. The models are similar in the sense that they both demand that the forager keep track of the direction to the nest entrance so that she may step toward it, in the current model, or exit through it, in the previous one. In any case the cognitive burdens of this sort of in-nest navigation are expected to be low since the ants can rely on chemical gradients (*Heyman et al., 2017*). However, in comparison to the previous model, the current one alleviates the forager's need to keep track of her unloading rate. Sensing the unloading rate, that is the change in crop load over time, requires some form of memory of past crop load values. Therefore, the new model presents a simpler mechanism by removing this computational and memory burden.

Other than its simplicity and its lower cognitive demands, this model is preferable over the previous one for its greater explanatory power. It manages to explain both the linear foraging frequency and the deepening of foragers' paths in the nest, implying that both of these trends result from the same set of rules. The deepening of foragers' visits with colony satiation may be a widespread phenomenon, as it was also observed in honeybees (*Seeley, 1989*).

Additionally, our analytical understanding of the described behavioral rules emphasises the robustness of the system to intrinsic forager parameters, such as threshold value and bias strengths. So long as there is a crop load at which the forager's movement bias switches from inward to outward, her exiting frequency is expected to be linear with her unloading rate (neglecting boundary effects). Since the forager's unloading rate is controlled by her recipients, her exit frequency is linked to the colony. This allows for different foragers to have different movement biases even within the same nest, and still the relationship between their exit frequency and colony satiation will remain intact.

On the other hand, the sensitivity of the forager's unloading rate to the crop loads of the ants she encounters means that a linear relationship between her foraging frequency and the total colony hunger requires her receivers to be representative of the colony. While our experiments indeed display a representative sample of receivers and a linear relationship with colony state, our model predicts that different interaction patterns will yield different results. In cases where the forager encounters non-representative subgroups of the colony, her foraging frequency is expected to be linear with the state of her sample, but this may no longer translate to linearity with the collective state of the colony. In nature, ants' nests are typically composed of multiple chambers (*Tschinkel, 2004*; *Tschinkel, 2005*; *Heyman et al., 2017*), thus the nest-ant distribution is more clustered and organized than in the artificial single-chamber nest that was used in our experiments (*Fard et al., 2020*). Accordingly, it may be that nest architecture will affect the sampling characteristics of the foragers, and consequently their foraging frequency (*Pinter-Wollman, 2015*; *Bidari and Kilpatrick, 2021*). Other factors that may affect the forager's sample include the number of nest entrances (*Lehue et al., 2020*; *Mitrus, 2021*),

the density of ants in the nest, and the topology of the colony's trophallactic network (*Sendova-Franks et al., 2010*, *Wikle et al., 2019*; *Quque et al., 2021*; *Bles et al., 2018*; *Planckaert et al., 2019*, *Mersch et al., 2013*; *Quevillon et al., 2015*; *Stroeymeyt et al., 2018*; *Alciatore et al., 2021*). Lastly, note that we describe the colony's feeding process after starvation. In nature such a state may occur when environmental conditions don't provide a stable supply of food. The level of hunger of the colony may very well affect the trophallactic network (*Sendova-Franks et al., 2010*). Fortunately, modern tracking methods enable to acquire more data on trophallactic networks to explore these potential effects (*Gernat et al., 2018*; *Baltiansky et al., 2021*).

Both the model presented here and those presented in our previous work relate foraging rates to forager decisions. However, the nature and timing of these decisions differ greatly. Although the distinction between these two alternatives is typically ignored, we would like to argue that there is a clear advantage to the model presented here. Taking a decision means choosing among the spectrum of affordances (*Stoffregen, 2003*), or currently available courses of action (*Budaev et al., 2019*). In the model presented here, upon reaching a certain crop load threshold, the forager decides to shift her bias toward the exit, a perfectly plausible decision. Then when her motion brings her to the nest entrance, she automatically exits. In the models presented in our previous work, the decision taken by the forager is a decision to exit. However, for an ant that is far into the nest, exiting is not an affordance. Even if the ant decides to cease interactions and exit she is not at the entrance. This ant must first traverse a highly dynamic and unpredictable environment wherein she may be exposed to further information or encounter nestmates that initiate further interactions. Any attempt to define the nature or timing of a decision that includes unavailable affordances is therefore liable to lead to inconsistencies and limits the models that use it.

The model presented here and those presented in our previous work were constructed to describe the same data. Indeed, when it comes to forager relating exit rates to colony state they yield highly similar results. However, this does not mean the decision to exit vs the decision to change bias models are indistinguishable. Rather, the models would predict observable behavioral differences. Since our previous models did not include any notion of space we can not directly compare them to our current more comprehensive model. However, experiments that differentiate between the two decision types can easily be envisioned. A decision model with predictive power should allow us to pinpoint, in real-time, the moment at which a forager takes her decision to exit. We could imagine an manipulation where, at this moment, a large number of hungry ants are added between the forager and the nest entrance. The previous models would predict that, since the decision has already been taken, this manipulation will not deter her from quickly reaching the entrance and exiting. The model presented in the current work would predict the opposite: as long as there are hungry ants around the forager will keep on interacting. Clearly, further differences between the models are to be expected if one goes beyond behavior and into the neuronal correlates of information accumulation and decision making processes.

We suggest to identify the mechanism presented in our current model as a very simple form of stigmergy (*Grasse, 1960*). Stigmergy is a means by which social insects coordinate their behaviors through alteration of the physical environment. Ants can, for example, alter the environment by leaving a pheromone mark on the ground or starting to dig a new tunnel and when other ants react to these environmental signals pheromone trails (*Theraulaz and Bonabeau, 1999*) or elaborate nest structures (*Theraulaz et al., 1998*) may emerge. Furthermore, it is appreciated that this form of emergence work to reduces the cognitive abilities required of the participating individuals. In our model, all a forager does is move within the nest. Clearly, since the ant is an embodied agent (*Wilson, 2002*), this motion alters the physical environment and can therefore be viewed as a simple form of stigmergy. Furthermore, the distance between the forager and nest entrance is a dynamic variable that integrates all her previous steps. This variable is simply where the ant is and therefore she is not required to compute or store it internally. Hence, similar to the more complex forms of stigmergy, motion relieves the forager's cognitive burden. Interestingly, this simple form of stigmergy can be employed by a single agent. A similar process, in which the location of a single ant relieves her from measuring interaction rates during quorum sensing was recently proposed by *Pavlic et al., 2021* . Similarly, the physical location of a cockroach (*Halloy et al., 2007*) or a fish (*Berdahl et al., 2013*) during collective shelter selection relieves them from remembering or even being aware of their choice. A similar mechanism, wherein the mean location of a group of ants provides a realization of the abstract cognitive

variable typically typically used in neuroscience decision models was recently demonstrated in ants (*Ayalon et al., 2021*).

In summary, the model we present here for foraging frequency regulation supports the same notion. If the movement of the foragers is neglected, it may seem that in order for foraging frequency regulation to emerge, foragers must accumulate information on their changing crop load in order to decide when to exit the nest (*Greenwald et al., 2018*). However, here we show that once forager movement is considered, the decision when to exit the nest is no longer an internal decision of the forager, but an external decision made by the collective that includes the forager, the colony, and their physical environment. Accordingly, the internal behavior of the forager then solely relies on her current crop load, with no need for her to accumulate information on its history. The cumulative information on past crop load values is represented by the forager's position in the nest and is thus stored externally in physical space. This exemplifies how utilization of an individual's position in space can reduce its cognitive demands without detracting from its computational contribution to group-level emergence.

## Materials and methods

### Experimental setup
The experiments used to conduct this research are those used in *Greenwald et al., 2018*.

### Data analysis
Data was analysed in Python using the following packages: Numpy (*Oliphant, 2006*), Matplotlib (*Hunter, 2007*), openCV (*Bradski, 2000*) and Pandas (*McKinney, 2010*).

### Agent-based models
The agent-based models are described according to the protocol laid out by *Grimm et al., 2006*; *Grimm et al., 2020*. Two models are presented, a two-dimensional model in continuous space and a 1-dimensional model in discrete space. Both models include a single forager which has two stochastic walking tendencies: the forager tends deeper into the nest when the amount of food in her crop is above a threshold value and tends toward the entrance once her crop level drops below the threshold. Models progress in the following way: the forager begins at the entrance with a full crop and walks through the nest, unloading food to each nest-ant she meets according to her trophallaxis rule. Once she has unloaded enough food, she switches walking tendency, and tends toward the entrance. Upon reaching the entrance, she refills and proceeds to re-enter the nest.

The model is written in Python with a GUI written in Java. Scheduling in the 1D model was carried out through a modified version of the mesa scheduling module (*Masad and Kazil, 2015*).

#### Purpose and patterns
The purpose of the model is to determine whether the three rules described in section 3.2 are sufficient to recapitulate the forager's exit frequency relation with colony hunger. Other patterns of the foragers' unloading bouts are used to determine the accuracy of the model, including depth, exiting crop state, unloading rate, and the state of their interaction partners.

#### State variables and scale
The model is comprised of individual agents representing ants; ants can be grouped into two sub-populations, foragers and nest-ants. These two populations are representative of what is seen in ant colonies in the scope of food dissemination. The model is also treated as an individual object to allow for data collection and parameter setting. Model parameters and their values are specified in *Table 2*.

In the 1D model, biases {a, b, c} are to be read as such; a is probability to step one cell outwards, b is probability to stay in the same cell, c is probability to step one cell inwards. In the 2D model, the forager moves 0.2 ant-lengths at every step (the average empirical velocity of foragers). After every interaction, she samples a new direction from a list of angles extracted empirically, given her crop load (*Appendix 1—figure 1*). Furthermore, nest-ants move 0.2 ant-lengths in a random direction at every step.

In both models one forager was initialized and simulations were run until the colony was sufficiently satiated.

**Table 2.** Parameter values for different groups of agents in both models. Parameters given to all agents are described under the 'Ants' sub-population.

| Sub-population | Parameter | Model | Value |
|---|---|---|---|
| Ants | | | |
| | Crop state capacity | All | 1 |
| | Movement speed | 2D | 0.2 ant-lengths second$^{-1}$ |
| Nest-ants | | | |
| | Initial crop state | All | 0 |
| | Position | | |
| | | 1D | 45 ants, one on every cell |
| | | 2D | 89 ants randomly placed |
| | Radius of interaction | 2D | 0.2 ant-lengths |
| Forager | | | |
| | Initial crop state | All | 1 |
| | Threshold value | All | 0.45 |
| | Initial position | All | Entrance of nest |
| | Foraging time | All | 0 |
| | Interaction proportion | All | $p \sim Exp(\frac{1}{0.15})$ |
| | Biases in state A | 1D | {0.32,0.22,0.46} |
| | Biases in state B | 1D | {0.53,0.31,0.16} |
| | Possible angles in state A | 2D | *Appendix 1—figure 1*, above |
| | Possible angles in state B | 2D | *Appendix 1—figure 1*, below |
| | Boarder reflection noise | 2D | [–0.3 radians, 0.3 radians] |

## 1D model

The nest length is 45 cells, plus 1 entrance cell. The entrance and deepest cell in the nest are reflecting boundaries, forcing the forager to step inwards/outwards, respectively, the step after it reaches said cell.

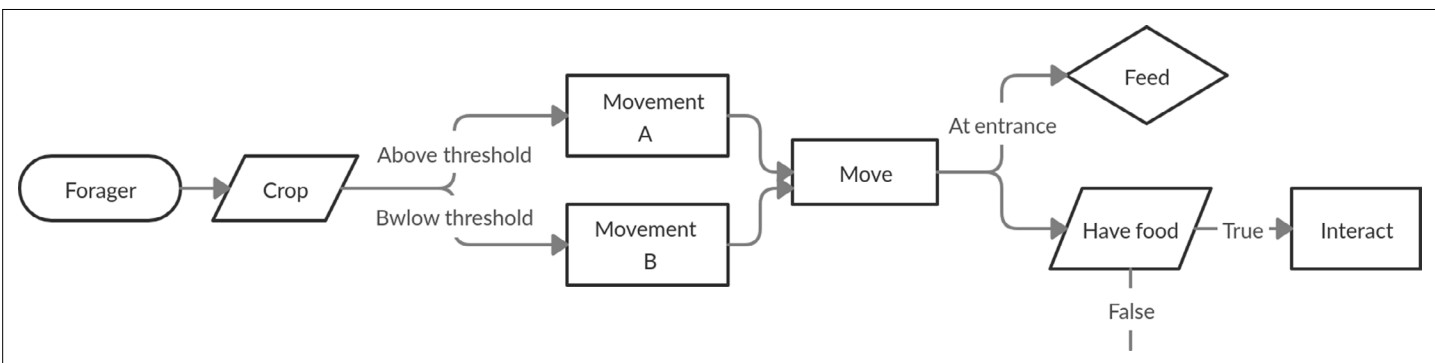

**Figure 6.** Schematic detailing the process of a forager at every step of the simulation. The forager first moves according to a crop-dependent movement rule. Then either feeds, if at the entrance, or attempts to interact with another agent.

## 2D model

The nest is of size 12 by 12 ant-lengths, Where the bottom left corner is the point of entrance/exit. When the forager reaches a nest boundary her direction is reflected with noise of 0.3 radians.

## Process overview and scheduling

The process of the forager in both models is described by the flow diagram in *Figure 6*.

Time in the model is discrete. In the 1D model, each step in the simulation represents the time it takes for the forager to step one ant length. In the 2D model, each step represents one second.

## Design concepts

- Emergence: Forager dynamics emerge from the behaviors of the model. Interactions and movement are hard-coded, however, dynamics such as duration, depth, exiting crop and colony state progression all develop only as a consequence of these behaviors. Hence, the foragers' adaptation to the changing colony state occurs implicitly via these behaviors and its position.
- Sensing: All agents are assumed to know their crop levels, and the forager also knows a movement threshold crop level. Agents are not assumed to know where they are in the nest or any information about other agents/the system.
- Interactions: Trophallaxis between agents is modeled explicitly.
- Stochasticity: Trophallaxis and movement are both modeled as stochastic behaviors.
- Observation: Simulations were repeated 200 times, with crop state of all agents at every step averaged over these repeats. Forager-specific data; crop, position, current duration in the nest, was recorded at every step for every individual run of the simulation. Interaction volumes and partners' states were recorded for every step in every individual run of the simulation. Unloading bout duration and exiting crop were recorded every time the forager returned to the entrance for every individual run of the simulation. In analyzing the results of the models, unloading bouts below a certain duration were omitted. Empirically, foragers were not observed to perform very short bouts, probably because they have a memory of entering the nest. This was captured by rejecting bouts shorter than 9 s in the 2D model, and shorter than three steps in the 1D model.

## Initialization and termination

Every simulation was initialized with empty nest-ants and a fed forager. This mimics the data collected from wet-lab experiments, in which colonies were starved for 1–2 weeks prior and data collection only began after the first time a forager leaves the nest to find food. The forager is initialized in the entrance, and the nest ants are initialized in random positions (2D model) or in every cell (1D model).

Simulations were terminated after all nest-ants were at least 95% full.

## Input

No external input into the models was used.

## Sub-models

- Trophallaxis rate: In the 1D model, the forager performs trophallaxis at every step with the ant in her cell. In the 2D model, the forager performs trophallaxis if there is a nest-ant within the radius of interaction.
- Trophallaxis volume: The forager transfers a fraction of the recipients empty crop space. The fraction is sampled from an exponential distribution with a mean of 0.15 (based on empirical data from *Greenwald et al., 2018*).

## Acknowledgements

We thank Efrat Greenwald for collecting the empirical data used in this study. Thanks to Amos Korman and Jean-Pierre Eckmann for mathematical consultation and ideas, and to Aviram Gellblum for coding advice. This research was supported by the Minerva Foundation, the Israel Science Foundation, Grant No. 1727/20, the European Research Council (ERC) under the European Unions Horizon 2020 research and innovation program (Grant agreements No. 648032 and 770964).

## Additional information

### Funding

| Funder | Grant reference number | Author |
|---|---|---|
| Minerva Foundation | | Ofer Feinerman |
| Israel Science Foundation | 1727/20 | Ofer Feinerman |
| Horizon 2020 Framework Programme | 648032 | Ofer Feinerman |
| Horizon 2020 Framework Programme | 770964 | Ofer Feinerman |

The funders had no role in study design, data collection and interpretation, or the decision to submit the work for publication.

### Author contributions

Lior Baltiansky, Conceptualization, Software, Formal analysis, Validation, Visualization, Methodology, Writing - original draft, Writing - review and editing; Guy Frankel, Conceptualization, Software, Formal analysis, Visualization, Methodology, Writing - original draft, Writing - review and editing; Ofer Feinerman, Conceptualization, Supervision, Funding acquisition, Validation, Visualization, Methodology, Writing - original draft, Writing - review and editing

### Author ORCIDs

Lior Baltiansky (iD) http://orcid.org/0000-0003-3870-1788
Ofer Feinerman (iD) http://orcid.org/0000-0003-4145-0238

### Decision letter and Author response

Decision letter https://doi.org/10.7554/eLife.77659.sa1
Author response https://doi.org/10.7554/eLife.77659.sa2

## Additional files

### Supplementary files
• Transparent reporting form

### Data availability

Figure 2 - Source Data 1 contains all experimental data used for the empirical analysis. Figure 4 - Source Data 1 and Figure 4 - Source Data 2 contain simulated data used for the analysis of the agent-based models.Python code for the agent-based model is available on GitHub.

The following dataset was generated:

| Author(s) | Year | Dataset title | Dataset URL | Database and Identifier |
|---|---|---|---|---|
| Frankel G, Baltiansky L, Feinerman O | 2022 | Food dissemination agent-based model | https://doi.org/10.5281/zenodo.7377670 | Zenodo, 10.5281/zenodo.7377670 |

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

# Appendix 1

## Foragers' movement directions

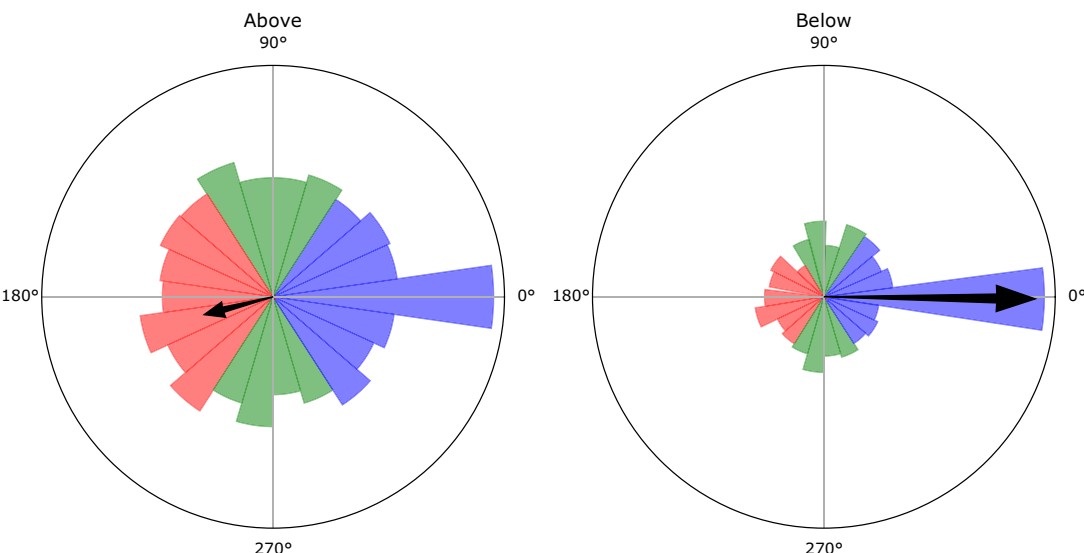

**Appendix 1—figure 1.** Empirical distributions of the angles between the foragers' direction of movement and the direction to the nest entrance (an angle of $0^o$ represents a direct movement toward the entrance). Black arrow represents the mean of the distribution. Data was sampled at the end of each interaction of a forager. The angles are presented in two distributions, one where the foragers' crop load was above the identified threshold (left), and one where the foragers' crop load was below this threshold (right). The threshold value was extracted from the data presented in *Figure 1* in the main text. Above the threshold foragers had a net bias away from the entrance (mean ± STD: 194.5° ± 133.3°), and below the threshold a net bias toward the entrance (mean ± STD: 359.4° ± 85.0°). In the continuous 2D simulation, the foragers' direction of movement was determined by randomly sampling an angle from the empirical angle distributions.

## Nest-ant movement in 1D model

To translate the empirical 2D movement of nest-ants to 1D for the 1D agent-based model, we mapped all nest-ants' 2D positions into a 1D ranking according to their Euclidean distance from the entrance. To quantify the degree of their movement between consecutive unloading bouts, we calculated the Kendall $\tau$ rank correlation index for each pair of consecutive sets of rankings. The coefficients for each one of our 3 experiments were distributed close to 0, indicating that the change in ants' rankings was close to what would be obtained by random shuffling. Indeed, we also randomly shuffled the empirical rankings for comparison, and the resulting coefficients were not different from those calculated on the non-shuffled empirical rankings (Figure *Appendix 1—figure 2*). Therefore, nest-ant movement in the 1D model was implemented as random shuffling between consecutive unloading bouts.

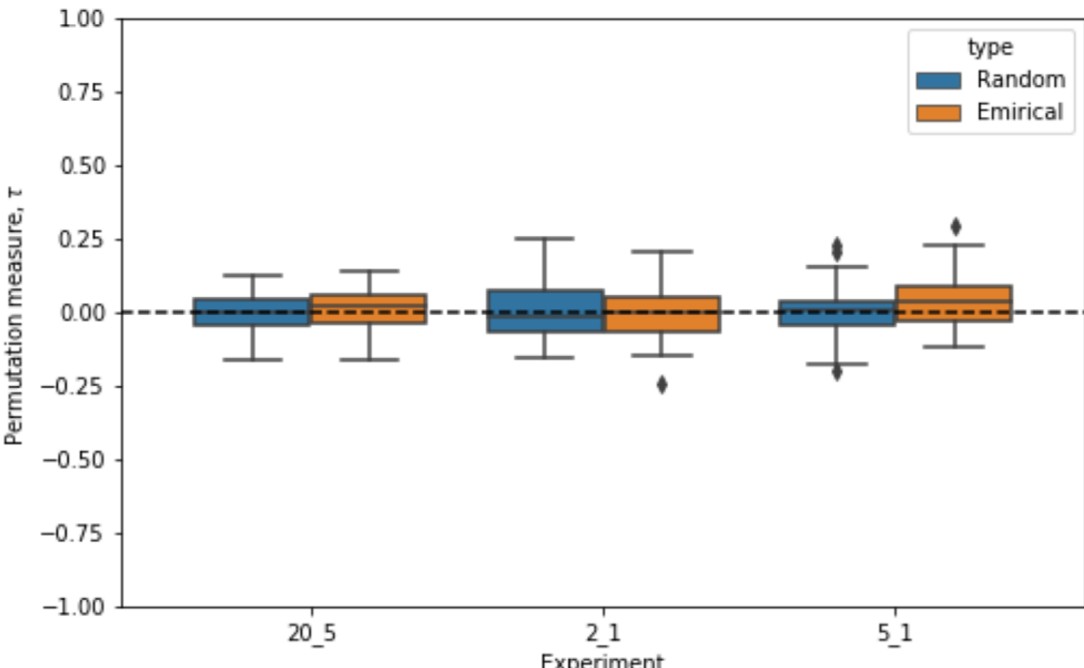

**Appendix 1—figure 2.** Distributions of the Kendall $\tau$ rank correlation coefficients for nest-ant movement in 3 experiments compared to those of fully random shuffling.

## Emergent linear relationship between foraging frequency and total colony hunger

In the main text, we have presented equations that explain the linear relationship between foraging frequency and total colony hunger for the extreme case, in which a forager walks only inward when her crop load exceeds the threshold, and only outward when it is below the threshold. Here we generalize these equations for the cases where the forager's bias is set with partial probabilities to walk in each direction.

Let us describe the forager's probabilistic walk as a random walk with a crop-load dependent bias $B(c)$, where $c$ is the forager's current crop load, and $B(c)$ is a set of 3 complementary fractions describing the probabilities to take one step inward, outward or to stay in the same cell, given $c$. In our case, a crop-load threshold $c^*$ defines 2 biases: $B(c > c^*)$ where the probability to step inward is greater than the probability to step outward, generating a net inward drift, and $B(c \leq c^*)$ where the probability to step outward is greater than the probability to step inward, generating a net outward drift. We will denote these biases $B_{in}$ and $B_{out}$, respectively. For example, the biases used in the agent-based model are presented in *Table 1*, the first row corresponding to $B_{out}$ and the second to $B_{in}$.

When compared to the extreme case presented in the main text, the main difference that this probabilistic walk introduces is that as the forager walks stochastically back and forth between the nest ants, she may interact multiple times with the same ants before switching her bias. For any biased random walker walking on an infinite line, the average number of times it steps on a specific position is a constant, the value of which depends on the value of the bias. This is due to the fact that a biased random walk is Markovian, such that the direction of the next step is independent of the walker's position. Therefore, we can separately analyze the two phases of the forager's walk in the nest: (1) when she enters the nest and drifts inwards with bias $B_{in}$ until she reaches her crop-load threshold, and (2) after she reaches the threshold and drifts back toward the nest entrance with bias $B_{out}$. During each one of those phases, the average number of times the forager interacts with each ant is a different constant, which we denote $s(B_{in})$ and $s(B_{out})$, respectively. Note that this holds under the assumption that the forager is walking far enough from the nest boundaries. The average number of encounters with ants that are close to the boundaries may depend on their position, but in any case should stay quite constant during the entire course of the feeding process, and therefore we neglect this complication for the sake of our analysis.

Similarly to the extreme case presented in the main text, the nest-ant movement rule dictates that the forager interacts with a representative sample of the colony at each unloading bout, such that the average state of the forager's partners is equal to the colony state, $F$. However, for the probabilistic walk, this is true only for the first time the forager interacts with this partner. When repeatedly giving food to the same ants, their crop state gradually increases, affecting the amount of food they will receive in each successive interaction. Since the trophallaxis rule states that the average amount of food delivered at each interaction is a fraction $\alpha$ of the receiver's empty crop space, the receiver's empty crop space decreases, on average, by a factor of $(1 - \alpha)$ at each successive interaction. Hence, treating the receiver's empty crop space as a geometric series with common ratio $(1 - \alpha)$, the total amount of food given to an ant with crop state $F$ after $N$ interactions is, on average:

$$\Delta c(N) = (1 - F) \sum_{n=1}^{N} \alpha(1 - \alpha)^{n-1} \tag{A1}$$

Next, looking at the average amount of interactions each ant holds during the forager's walk, $s = \langle N \rangle$, the total amount of food given to each ant can be approximated by (note that we use the integral approximation of the above sum as $s$ may not be an integer):

$$\langle \Delta c_{ant} \rangle \approx (1 - F) \cdot \int_{n=1}^{s} \alpha(1 - \alpha)^{n-1} dn \tag{A2}$$

Now, the average position at which the forager reaches her threshold can be expressed as the average number of unique ants she interacts with before unloading 1-$c^*$ of her crop. For the probabilistic case, this equals:

$$\langle x_{switch} \rangle = \frac{1 - c^*}{\Delta c_{ant}(B_{in})} = \frac{1 - c^*}{(1 - F) \cdot \int_{n=1}^{s(B_{in})} \alpha(1 - \alpha)^{n-1} dn} \tag{A3}$$

Since $s(B_{in})$ and $\alpha$ are constants, $\int_{n=1}^{s(B_{in})} \alpha(1 - \alpha)^{n-1} dn$ is also a constant, which we denote $\eta$, and obtain:

$$\langle x_{switch} \rangle = \frac{1 - c^*}{(1 - F) \cdot \eta} \tag{A4}$$

As in the extreme case, the average duration of a forager's unloading bout in the nest is the average time it takes her to reach $\langle x_{switch} \rangle$ from the entrance plus the average time it takes her to reach the entrance back from $x_{switch}$. For the probabilistic case, these may be expressed as $\langle x_{switch} \rangle \cdot s(B_{in})$ and $\langle x_{switch} \rangle \cdot s(B_{out})$, respectively. Therefore, the average time the forager spends in the nest before exiting is:

$$\langle T \rangle = \langle x_{switch} \rangle \cdot (s(B_{in}) + s(B_{out})) = \frac{(1 - c^*)(s(B_{in}) + s(B_{out}))}{(1 - F) \cdot \eta} \tag{A5}$$

The exiting frequency is thus:

$$R = \frac{1}{\langle T \rangle} = \frac{(1 - F) \cdot \eta}{(1 - c^*)(s(B_{in}) + s(B_{out}))} = (1 - F) \cdot \gamma \tag{A6}$$

where $\gamma = \frac{\eta}{(1 - c^*)(s(B_{in}) + s(B_{out}))}$ is a constant.

Hence, we get a foraging frequency $R$ which is linear with colony hunger $(1 - F)$ for the probabilistic case as well.

Plugging in the values for the constants in *Equation A6*, verifies that $\gamma$ matches the slope of the output of the 1D simulation (*Figure 5*). The constants that compose the factor $\gamma$ are:

- $c^* = 0.45$: the forager's crop load threshold, set based on empirical observation
- $\alpha = 0.15$: the average amount of food transferred in an interaction, in terms of fraction of the receiver's empty crop space, set based on empirical observation
- $s(B_{in}) = 2.2$: the average number of times a biased random walker is expected to visit each position for the inward bias, estimated as explained below
- $s(B_{out}) = 1.8$: the average number of times a biased random walker is expected to visit each position for the outward bias, estimated as explained below

The latter two constants were estimated by treating the system as an absorbing Markov chain, where the entrance to the nest is an absorbing state, and the rest of the positions are transient states. The fundamental matrix of this chain is $N = (I - Q)^{-1}$, where $Q$ is the transition matrix of the transient

states. Then, starting at the nest entrance, the expected number of steps on position $i$ before being absorbed back at the entrance is $N_{1,i}$. Setting the transition matrix $Q$ according to the defined inward and outward biases (**Table 1**), we obtain the expected values of $s(B_{in}) = 2.2$ and $s(B_{out}) = 1.8$. Note that this approach assumes a semi-infinite nest, thus the obtained values are approximations that neglect boundary effects.

Altogether, these constants yield $\gamma = 0.074$. This value is consistent with the slope obtained by a linear fit to the simulated data of exit frequency vs. empty colony state ($slope = 0.073$).

