## [Editor Report]

This valuable study is of relevance to the field of collective animal behaviour. The proposed crop-cue-based motion-switching rules provide a welcome alternative to other models that assume far more deliberative abilities of ants. The authors present solid evidence to back up their claims.

---

## [Decision Letter]

**Decision letter after peer review:**

Thank you for submitting your article "Emergent regulation of ant foraging frequency through a computationally inexpensive forager movement rule" for consideration by *eLife*.

Your article has been reviewed by three peer reviewers, and the evaluation has been overseen by a Reviewing Editor and Christian Rutz as the Senior Editor. The following individual involved in the review of your submission has agreed to reveal their identity: Theodore P. Pavlic (Reviewer #2).

The reviewers have discussed their reviews with one another, and the Reviewing Editor has drafted this decision letter to help you prepare a revised submission.

Essential revisions:

1) It was confusing to the reviewers how exactly the inward/outward directions were defined. Is it simply away, or towards the entrance? It is not clear from the text, and since this system is not symmetric (cubic with entrance at one of the corners) the authors should clarify this point.

2) For the biased random walk analysis of the ants, the authors "coarse-grained" the steps as being "inwards", "outwards" and "stay". It is not clear how this level of granulation is justified. Since the authors have access to the actual trajectories and all trophallaxis events, why not just calculate the actual turning angles between consecutive steps the ants take? This would give an actual assessment of both the bias and the noise imposed on the random walks, which the authors could then use directly in their models. Some discussion of this point is important.

3) It would be important to better connect the authors' previous mechanism (relating the colony's response to individual ants sensing their own food levels and its temporal dynamics) to the new mechanism (spatial-temporal dynamics). Are they mutually exclusive? It would be useful to elaborate on this in the Discussion.

4) The addition of a few supplementary movies from the experiments, showing ants moving toward the entrance with low food loads, and moving away from the entrance with high food loads, would be extremely helpful.

5) Much of the authors' argument rests on trajectories and statistics generated from a two-dimensional computational simulation that may be overly simplistic. The computational model simulates a single forager (as opposed to multiple foragers) arriving to a nest that is partitioned into a grid of squares with an immobile ant in the center of every square. Foragers move in discrete steps from square to square, with the guarantee of an interaction in each step. This "grid world" model of ant nest movements is significantly different from the experience of real foraging ants returning to the nest, and the authors even admit that deviations between the empirical data and the computational model may be due to nest-ant clumping and interaction sparsity in the paths of real ants. Continuous-motion agent-based models are commonly used to investigate collective-motion hypotheses, and so the choice of a grid world model instead seems surprising and weakens the authors' arguments. Furthermore, while the deterministic mathematical model of grid-world forager trajectories seems overly simplistic, the stochastic model in the Appendix that attempts to validate the deterministic model's results seems to have some potential flaws and is itself not validated experimentally against replicated simulation data. Instead of perfecting these models, the authors could bolster their arguments using more familiar approaches from statistical mechanics that might help explain the likely depth an ant "diffuses" into such a nest. In the current form of the manuscript, the mathematical models do not add much beyond the simulation models (and the lack of replication of the simulated data may make some readers wonder if the example trajectories were representative). Further discussion of continuum models would help to bolster the authors' claims, and the reviewers agreed that direct comparison of the authors' results from grid-based simulations to simulations from continuum models likely would be the most effective way to strengthen the manuscript and support its conclusions (see comments from Reviewer #2 for more details).

6) There are a few questionable parameters that the authors have chosen in their model, likely for analytical tractability. For example, the authors assume that at each interaction between a forager and a nest ant, the forager offloads enough food to fill 15% of the crop space remaining in the receiving ant. One can assume that this parameter is something like the 63.21% associated with an exponential time constant or may be based on empirical measurements of transfer in real ants, but the actual justification is not completely clear from the manuscript. Because the mathematical models make predictions that depend on these parameters, their existence (and plausible values) is itself an important assumption that needs to be defended for the argument to be compelling.

7) The behavioral model described by the authors assumes that ants are able to choose a direction toward their nest's entrance at any time. This within-nest path-integration ability does not seem cognitively inexpensive, which narrows the cognitive distance between the behavioral model they propose here and the one they had proposed in their prior work, and weakens the argument for the relevance of this new model. The authors failed to place their work within the context of other simple cue-based motion-switching behaviors discussed in the literature for other taxa -- such as "running" and "tumbling" in *E. coli* bacteria -- but if they had, they might have envisioned an alternative crop-based motion rule that would have the same effect as their current rule (i.e., movement toward the entrance on low crop state) without having to assert that the ant moves directly back towards the entrance. Bridging the work to these other studies would be important here.

8) Focusing on the explanatory power of this model specifically for (some) ants, the authors do not address how to empirically reconcile the ambiguity between the more cognitive mechanisms proposed in their previous work (where ants "decide" to exit a nest) and the current proposal (where the nest cavity "decides" when the ant will exit). For this new hypothesis to be useful, it must be empirically discriminable from the previous hypothesis. At first glance, it is difficult to imagine an experiment that would lead to different predicted behavior from the two different hypotheses. In other words, at the moment, it seems impossible to tell whether the "ant decide" or the "nest decide" model is a better predictor of real ant behavior/cognitive architectures. The lack of discriminability becomes even more problematic when considering that the current version of the model actually increases some cognitive demands by assuming (as described above) that ants keep track of the position of the entrance over the trajectory within the nest.

9) In the stochastic model in the Appendix (an integral is used when instead of a sum, perhaps?), it seems like the average values s(Bin) and s(Bout) should depend on F. However, they are treated as constants in Equations (S3) - (S5). If the authors tried to empirically validate the stochastic model by generating many simulated replications and then plotting averages against this prediction, they would likely have a hard time calculating s(Bin) and s(Bout) to generate their numerical predictions. The authors should clarify this point.

10) "It has recently been suggested that physical space can be utilised to offload computation from individuals' cognition to their environment in the context of collective quorum sensing ([5])." It seems surprising to say that this is the first time this has been suggested. For example, the literature on the effects of nest architecture cited by Reviewer #3 is based on this idea (see below). More generally, the idea that movement patterns determine encounter rates and thus communication was suggested for ants decades ago. This study seems to sidestep many spatially explicit models on information exchange through encounters (e.g., see review in Gordon 2020 Ann Ent Soc 2020doi: 10.1093/aesa/saaa03). The models use assumptions that ignore the effects of space. The impact of these assumptions should at least be considered.

11) The manuscript says nothing about the empirical data which were obtained in another study. This manuscript should say how "average crop load" was measured, including some measure of variation. The manuscript should also say how "foraging frequency" was measured and under what conditions. How often are these conditions likely to occur for colonies of this species?

12) What is the "linearity of foraging frequency"? Line 49: "Progress has also been made toward revealing the local mechanism underlying the linearity of foraging frequency, though to a lesser extent. " Again, on lines 105-106: "emergent linear relationship between foraging frequency and total colony hunger." Better definition of this term is important.

13) Figure 5D – empirical results: Why does the forager have a higher crop load at the end of its time inside the nest than at the beginning?

14) If "hunger" is defined as below as the amount of food in the colony, then it is circular (not "intriguing") to say that the rate at which food comes in matches the total level of "hunger" or level of food.

15) It seems strange to cite Oster and Wilson to say that ants collecting food are called foragers. There is at least 100 years of work on ants before Oster and Wilson that referred to "foragers".

16) Lines 36-38: "Trophallaxis is the main food-sharing method in many ant species 36 ([12]). Each time a laden forager returns to the nest, she unloads the food from her crop to several receivers via 37 trophallaxis. The food further circulates through a complex trophallactic network among all colony members38 ([9], [13]-[17])." This is a misleading way of framing this study, because it equates the distribution of food sources among ants within the nest with the unloading of nectar by foragers. There are many species that use trophallaxis but not directly from foragers.

17) The results suggest that unloading is associated with whether a forager moves toward or away from the nest entrance. This is called the "deeper nest", but it seems the previous empirical study was performed in a flat arena, and the simulations do not include anything about depth. Thus it gives the impression that an ant associates unloading with going up or down, but in this study, unloading was associated with toward or away from the entrance from an arena. It would be better not to use "deeper" to mean "away from the entrance" as this evokes an image of depth in an ant nest, which is misleading. Since "deep" has an ambiguous meaning here, it is difficult for the reader to know what "deepening" and "lengthening" mean in line 140: "The simulation qualitatively reproduced the lengthening and deepening of foragers' trips."

18) It was unclear what was meant by: "Note that contrary to the assumptions used in our previous paper ([1]), here a forager never decides to exit the nest. Rather, an exit occurs if the forager's motion brings her to the nest exit." What is the difference between 1) a decision to go to the nest exit and leave the nest, and 2) going to the nest exit and leaving? In the literature on behaviour, decision-making, and cognition, 1) and 2) are the same.

*Reviewer #2 (Recommendations for the authors):*

This very clever manuscript was a joy to read, and I look forward to when it is finally published. These crop-cue-based motion-switching rules provide a welcome alternative to other models that assume far more deliberative abilities on ants, and it will be valuable to add this example to the collective motion and collective decision-making literature. That said, I think there are three major issues that I feel warrant addressing in a revised version: overly simplistic models, no connections to similar phenomena in motion ecology as well as statistical mechanics, and potential flaws in the stochastic modeling approach. I will address each of these below.

Issue 1: Overly simplistic models

The manuscript's arguments are currently tailored to overly simplistic models. Choosing models for natural systems necessarily means leaving out some realism, but the grid-world models used by the current manuscript do not achieve the appropriate benefit-cost balance of analytical tractability to organismal fidelity. A good, illustrative simulation model need not have all of the details of the real system but it should have plausible relative scaling. A grid-world model of an ant nest, where there is an ant on every square and the single incoming forager moves from every ant to every other ant at each step is a significantly distorted proxy for a real ant colony. Agent-based modeling tools (and/or API's) allow for quickly building models of mobile agents that move in an approximation of continuous space, where an incoming forager could be moving around a nest that itself had nestmates that were moving. Putting both types of agents into motion will create natural gaps and clumps that help to create a more realistic temporal scales of events – possibly allowing for Figures 4B and 4C to have the same units as Figure 4A. Furthermore, simulating multiple foragers simultaneously might be important as the foragers will effectively compete for off-loading opportunities. Although using a continuous-time model may seem to complicate building mathematical models, the more realistic motion rules may actually simplify some of the analysis as they can lead to justifying well-mixedness assumptions that allow for using mean-field ODE models. In summary, although the grid-world simulations provide interesting visual evidence that such a model can generate hunger-dependent penetration depths inside a colony, such grid-world models are not convincing when discussing the actual temporal duration of those trajectories. Demonstrating these results in continuous-time agent-based models with potentially multiple returning foragers as well as mobile nest ants will be convincing and will able to be scrutinized in terms of temporal fidelity as well.

Issue 2: Connections to motion ecology and statistical mechanics

The manuscript in its current form describes what would happen if an ant had the ability to decide whether to move deeper within the nest or turn around and move directly toward the exit. From a mechanistic perspective, it would make more sense to suggest a mechanism (or family of mechanisms) that *tend* to have those two effects without assuming that the ant can achieve both of those subtasks. For example, following the flocking literature, it seems much more likely that ants would be able to move "toward center" or "away from center" (or even "toward darkness" (skototactic) and "toward light" (phototactic)). If the cue-based switching proposed lead to these two outcomes and then ants could follow walls when unable to move further away from center, then it seems likely that the same tendencies identified by the authors would be met without actually having to assert that ants can path integrate an "entrance vector" continuously. So I would recommend re-running simulations with more generic "inward" and "outward" motions. It is my guess that a wide variety of switching behaviors will lead to similar outcomes (albeit with a lengthening of the duration an ant spends in a nest, which might actually bring the simulations closer to the real traces anyway).

Event-based switching from one searching behavior to another is not unprecedented in the motion literature. Fish schooling literature (from Iain Couzin et al.) has shown that switching from one velocity to another based on whether you're in a dark or light area can lead to aggregations of fish, for example. A wide variety of animals (and even ants, such as Temnothorax albipennis when searching for its lost leader in a tandem run) incorporate switches from straight runs to circular searching and back again based on cues. Plume tracking in many flying insects is thought to involve simple switching rules that help ensure movement "upstream" despite the ugly turbulent flows in the odor plumes that are far from a smooth gradient. And that brings me to the example I mentioned in the public review -- *E. coli* "running" and "tumbling," which has been associated with chemotactic gradient climbing. Interestingly, *E. coli* are too small to sense a spatial gradient, and so some sequential sampling is apparently incorporated to estimate when it is ready to switch from rotating flagella in one direction ("running" straight) to the other direction ("tumbling" randomly). That implies that even bacteria can sense rate, which is possibly an argument for ants being able to sense the rate that their crop is being depleted. That said, if we forget about using the bacteria as a minimal model of cognition, we can focus on "running" and "tumbling" as a motion framework that ants could be using too. If the hypothesized ants can be conceptualized as "running" at high crop state and "tumbling" at low crop state, then could they be interpreted as climbing a nutrient gradient (i.e., in toward the nest when the nest is full of food but out of the nest when the nest is not full of food)? Not only would generic "tumbling" (as opposed to "moving toward the entrance") be less cognitively demanding for ants, but making the connection between ant and bacterial motion rules would help extend the scope and scale of the potential impact of this manuscript. So I would encourage: (a) seeing if simply increasing the probability of making random turns when the crop is low leads to a similar result as the current approach, and (b) considering whether "run" and "tumble" provides a gradient-climbing interpretation of what the foraging ants might be doing (i.e., they either climb into a "full" nest or they climb out of an "empty" nest toward a full environment).

Along those lines, there seem to be significant missed opportunities to interpret the trajectory density from a statistical mechanics perspective. Stating that trajectories tend to penetrate deeper into a test when colonies are "full" and is shallower when colonies are "empty" suggests that colony state might be viewed as a kind of "temperature", and the depth of penetration could reflect a corresponding Boltzmann distribution setup by the motion of foraging ants diffusing into the colony -- where those foraging ants would be excited by the "temperature" of the colony. If this interpretation is correct, then this statistical-mechanics perspective suggests other mathematical models that would be more general and more convincing than the simplistic mathematical models within the current manuscript (see more comments about these below). Alternatively, it might be possible to think of a sort of "contact potential" between the foragers (from outside) and the nest ants. When the colony is full, foragers can diffuse very far into the nest before the "charge imbalance" stops them from going further. However, when the colony is hungry, the "charge imbalance" balances at a much shorter distance (and so there is very little diffusion). At this moment, these are just descriptive models which may fit the data well. However, these descriptive models have specific physical phenomena associated with them which may inspire other ways to think about the motion of the individual ants. In general, diffusion is a very fundamental process which certainly applies to ants moving randomly from place to place, and so it seems like the clear modulation of penetration depth by hunger state is very likely to represent a kind of temperature. In this interpretation, the fuller the ant colony, the "more energetic" the forager, which is a happy coincidence.

Issue 3: Possible flaws in stochastic modelling approach

In the stochastic model in the appendix (where an integral is used when I think a sum was intended), it seems like the average values s(Bin) and s(Bout) should depend on F. However, they are treated as constants in Equations (S3) – (S5). Consequently, the stochastic model doesn't make sense to me. If the authors tried to empirically validate the stochastic model by generating many simulated replications and then plotting averages against this prediction, I think they would have a hard time calculating s(Bin) and s(Bout) to generate their numerical predictions. If I were building this model, I would have probably started with Markov renewal-reward theory. The individual forager encounters ants randomly and exchanges a random amount of food with them. The renewal process counts up the number of encountered ants, and the reward is the accumulated amount of food transferred to other ants. Framed this way, a wide range of results on Markov renewal-reward processes can be used to characterize the experience of the forager.

An alternative approach to the stochastic modeling would be to consider the hitting time of a drift-diffusion process. The manuscript already discusses how the "hunger" of the colony tunes the drift of such a process, with a "full" colony creating significant drift away from the absorbing barrier and an "empty" colony creating significant drift toward the absorbing barrier. Why not actually try to model the ant formally this way and import all of the mathematics already developed for such a system?

*Reviewer #3 (Recommendations for the authors):*

Methodological questions:

1). Space

"It has recently been suggested that physical space can be utilised to offload computation from individuals' cognition to their environment in the context of collective quorum sensing ([5])." It seems strange to say that this is the first time this has been suggested. For example, the literature on the effects of nest architecture cited here is based on this idea.

More generally, the idea that movement patterns determine encounter rates and thus communication was suggested for ants decades ago. This study sidesteps many spatially explicit models on information exchange through encounters (e.g. review in Gordon 2020 Ann Ent Soc 2020doi: 10.1093/aesa/saaa03).

The models use assumptions that ignore the effects of space. The impact of these assumptions should at least be considered.

a. How does the crop load of a recipient influence its location inside the nest? Social insect colonies are spatially organized; e.g.:

Franks NR, Tofts C. Anim. Behav. doi:10.1006/anbe.1994.1261;

Mersch DP et al. 2013 doi:10.1126/science.1234316;

Crall et al. Nat. Comm. 9:1-13.

b. How does a forager's movement influence the probability of meeting another individual with a particular crop load?

Davidson 2017 J. R. Soc. Interface.http://doi.org/10.1098/rsif.2017.0413

The model considers only one meeting. How would a 2nd, 3rd, … encounter influence the results?

c. Setting the bias to go toward the entrance equal to the bias to move away also has a strong effect on the results. What is the effect of removing this assumption?

d. Variation among ants met in crop load determines the probability that a forager will encounter a particular set of crop loads for a particular movement pattern. E.g. O'Shea-Wheller et al. 2017. Proc. R. Soc. B Biol. Sci. 284: 20162237.

Assuming there is no variation probably has a strong effect on this result, lines 216-18: "Nevertheless, it turns out that the average amount of food given to each nest-ant is still proportional to (1 − F), and that since both the inward and outward biases are constant, the number of steps spent with each nest ant is, on average, also constant (neglecting boundary effects)."

2). Data

The manuscript says nothing about the empirical data which were obtained in another study. This manuscript should say how 'average crop load' was measured, including some measure of variation. The manuscript should also say how 'foraging frequency' was measured and under what conditions. How often are these conditions likely to occur for colonies of this species?

3). Linearity

What is the 'linearity of foraging frequency'?

Line 49: "Progress has also been made toward revealing the local mechanism underlying the linearity of foraging frequency, though to a lesser extent."

Again, lines 105-106: “emergent linear relationship between foraging frequency and total colony hunger.”

I think this is the relation of rate of foragers exiting the nest vs estimate of total amount of food in the crops of workers in the nest? Why is it important that this relationship be linear? It seems more likely that it would be nonlinear, e.g. that foragers would be more likely to exit when levels are very low and not as much when levels are high.

4). Unloading

Figure 5D – empirical results: Why does the forager have a higher crop load at the end of its time inside the nest than at the beginning?

Conceptual issues and presentation:

The manuscript refers to 'ant colonies', in the abstract, introduction and discussion, as if these results apply to all ant colonies. However, the species studied here is one that feeds on nectar. While there are many other such species, they are not the majority of ant species. What is unusual is that they ingest their food, instead of just carrying it back to the nest, and they must unload it before they can collect more. The manuscript should make it clear that the process described here has evolved in relation to this particular, unusual type of feeding. In fact a similar process has evolved independently in honey bees, which also collect nectar. While Seeley's work on this in honeybees (reference 44) is mentioned in passing, the manuscript does not discuss this resemblance between this aspect of foraging behavior in honey bees and a similar and relatively unusual one in ants.

1). If 'hunger' is defined as below as the amount of food in the colony, then it is circular (not 'intriguing') to say that the rate at which food comes in matches the total level of 'hunger' or level of food.

Line 31: "Intriguingly, the rate at which food enters the colony matches the total level of hunger in the colony 31 ([1], [8], [9]).

Line 43-44: "Specifically, each forager's unloading rate was proportional to the 4total "empty crop space" in the colony (hereinafter, 'colony hunger')."

2). Strange to cite Oster and Wilson to say that ants collecting food are called foragers. There is at least 100 years of work on ants before Oster and Wilson that referred to 'foragers'.

3). Line 31 – what is a distributed nature?

4). Trophallaxis

Lines 36-38y: "Trophallaxis is the main food-sharing method in many ant species 36 ([12]). Each time a laden forager returns to the nest, she unloads the food from her crop to several receivers via 37 trophallaxis. The food further circulates through a complex trophallactic network among all colony members38 ([9], [13]-[17])." This is a misleading way of framing this study because it equates the distribution of food sources among ants within the nest with the unloading of nectar by foragers. There are many species that use trophallaxis but not directly from foragers.

5). The results suggest that unloading is associated with whether a forager moves toward or away from the nest entrance. This is called the 'deeper nest', but it seems the previous empirical study was performed in a flat arena and the simulations do not have anything about depth. Thus it gives the impression that an ant associates unloading with going up or down, but in this study, unloading was associated with toward or away from the entrance from an arena. It would be better not to use 'deeper' to mean 'away from the entrance' as this evokes an image of depth in an ant nest, which is misleading. Since 'deep' has an ambiguous meaning here, it's difficult for the reader to know what 'deepening' and 'lengthening' mean in Line 140: "The simulation qualitatively reproduced the lengthening and deepening of foragers' trips."

6). This is puzzling: "Note that contrary to the assumptions used in our previous paper ([1]), here a forager never decides to exit the nest. Rather, an exit occurs if the forager's motion brings her to the nest exit." What is the difference between 1) a decision to go to the nest exit and leave the nest, and 2) going to the nest exit and leaving? In the literature on behavior, decision-making, and cognition, 1) and 2) are the same.

7) line 122: It will be confusing to readers to call the forager's moving around inside the nest a 'trip', since it is very common to call its journey outside the nest a 'trip'.

---

## [Author Response]

Essential revisions:(1) It was confusing to the reviewers how exactly the inward/outward directions were defined. Is it simply away, or towards the entrance? It is not clear from the text, and since this system is not symmetric (cubic with entrance at one of the corners) the authors should clarify this point.

The definitions of the inward/outward directions appear in lines 93-95:

“The probabilities of her next interaction to be farther from the entrance (step inward), closer to the entrance (step outward) or at the same distance from the entrance (stay), were calculated as a function of the forager’s crop state at the end of the interaction.”

The exact calculation of the probabilities is detailed in the caption of Figure 2, with a visualization in Figure 2C. We now also clarify in line 108 that by the term “depth” we mean distance from the entrance.

(2) For the biased random walk analysis of the ants, the authors "coarse-grained" the steps as being "inwards", "outwards" and "stay". It is not clear how this level of granulation is justified. Since the authors have access to the actual trajectories and all trophallaxis events, why not just calculate the actual turning angles between consecutive steps the ants take? This would give an actual assessment of both the bias and the noise imposed on the random walks, which the authors could then use directly in their models. Some discussion of this point is important.

We agree that the actual turning angles can be useful for a more precise description of the foragers’ movement, and a more realistic movement model in 2D. Per the reviewers’ suggestion, we now address these angles in the SI (Figure S1) and use them in our new 2D model. See changes in lines 129-131, 138-146.

While the precise turning angles lend themselves to analysis and simulation in 2D, we still view the “inward”/”outward” granulation as the insightful level of analysis. The coarse-grained analysis of “inwards”, “outwards” and “stay” is what revealed the crop-dependent pattern of motion. The probabilities of walking in these directions clearly depend on the crop-load of the forager – highlighting the threshold that separates between a net drift into the nest and a net drift out of the nest. This point is now explicit in lines 95-98.

The essence of the described regulation is that foragers enter and exit the nest at a rate that matches the colony’s needs. Therefore a mechanism that drives the forager inward when she has a lot of food and toward the exit when she has little food (regardless of the precise angles) is the appropriate simplification for understanding the emergent regulation. It is what allowed us to model the system as a 1D nest, a model which is analytically tractable and explains the observed emergence. This point is now made clear in lines 185-187.

(3) It would be important to better connect the authors' previous mechanism (relating the colony's response to individual ants sensing their own food levels and its temporal dynamics) to the new mechanism (spatial-temporal dynamics). Are they mutually exclusive? It would be useful to elaborate on this in the Discussion.

The answer depends on the degree of detail with which the data is observed.

Zooming out, if we would like to understand the rate at which foragers exit the nest then both models would, to a very large extent, agree. The models were constructed to describe the same data and are not easily distinguishable.

However, if we zoom in a bit closer and want to understand not only the time the forager exits the nest but her entire trajectory within the nest then the new model explains this while the old one does not attempt to. However, even if our old models did have some notion of space we expect that at this level of detail the models would be mutually exclusive. This is because the nature of decision the forager takes in both models is different. In the old model, at some point (regardless of her location within the nest) the forager decides to cease further interactions and exit. In the new model, the decision the forager takes is to change her characteristics of motion (and increase her bias towards the door). The two models would strongly disagree on what happens between the time of decision and the time of actual exit.

Zooming in even further, if we had the ability to measure and decipher activity within the ants brain then the computations the ant is making, the type of decision she takes, and the timing of decision, certainly make these models mutually exclusive.

We now explain these points in the new paragraph 9 of the discussion where we also suggest an experiment which would distinguish the two models. (lines 354-366)

(4) The addition of a few supplementary movies from the experiments, showing ants moving toward the entrance with low food loads, and moving away from the entrance with high food loads, would be extremely helpful.

We have now added two supplementary movies to Figure 2. One of them is early in the feeding process where the forager changes her bias nest to the entrance. The other is when the colony is closer to satiation and the forager changes her bias much later in the bout and much deeper within the nest.

(5) Much of the authors' argument rests on trajectories and statistics generated from a two-dimensional computational simulation that may be overly simplistic. The computational model simulates a single forager (as opposed to multiple foragers) arriving to a nest that is partitioned into a grid of squares with an immobile ant in the center of every square. Foragers move in discrete steps from square to square, with the guarantee of an interaction in each step. This "grid world" model of ant nest movements is significantly different from the experience of real foraging ants returning to the nest, and the authors even admit that deviations between the empirical data and the computational model may be due to nest-ant clumping and interaction sparsity in the paths of real ants. Continuous-motion agent-based models are commonly used to investigate collective-motion hypotheses, and so the choice of a grid world model instead seems surprising and weakens the authors' arguments. Furthermore, while the deterministic mathematical model of grid-world forager trajectories seems overly simplistic, the stochastic model in the Appendix that attempts to validate the deterministic model's results seems to have some potential flaws and is itself not validated experimentally against replicated simulation data. Instead of perfecting these models, the authors could bolster their arguments using more familiar approaches from statistical mechanics that might help explain the likely depth an ant "diffuses" into such a nest. In the current form of the manuscript, the mathematical models do not add much beyond the simulation models (and the lack of replication of the simulated data may make some readers wonder if the example trajectories were representative). Further discussion of continuum models would help to bolster the authors' claims,

We thank the reviewers for pointing out that the discrete model may weaken our manuscript. We now present a continuous-motion model instead. This model is more realistic than the previous one in that the ants’ movement is continuous, the nest is more sparse, and trophallaxis only occurs when the moving forager “meets” a moving ant.

Indeed, one may choose to analyze this rich system from a vast variety of models, such as diffusion, run-and-tumble, etc. Each approach could be fascinating and insightful on its own. Our goal was to understand the emergence of the regulation of foraging frequency, and our simple model was sufficient to explain it. Other modeling approaches are indeed very interesting, and in fact are currently being explored in another ongoing work of ours using similar experiments with larger colonies (which are closer to the continuum limit).

We wish to point out that our simulation data is from 200 replicates (in this version of the manuscript and the previous one). The trajectories in Figure 3 are examples from a single run for visualization, while Figures 4 and 5 show summary results from all runs.We are sorry that this information was missed, making our examples seem less credible. Information on replicates was previously mentioned in the caption of Figure 4, and is now reiterated in line 158 and in the caption of Figure 5 as well.

The questions raised regarding the potential flaws in our stochastic model are addressed in comment 9 below. Here let us mention that we added a paragraph that shows how the value of the slope predicted by our equations is verified by the slope that resulted from the replicated simulation data in Figure 4.

and the reviewers agreed that direct comparison of the authors' results from grid-based simulations to simulations from continuum models likely would be the most effective way to strengthen the manuscript and support its conclusions (see comments from Reviewer #2 for more details).

We have fully complied with the reviewers suggestion to construct and analyze a continuous 2D model of the unloading bouts. We have shown how this model qualitatively reproduces the sought after linear scaling between foraging frequency and total colony hunger (see figure 4) as well as other relations between forager behavior and collective colony states (see figure 5). Please note that although the continuous 2D model is more realistic than the previous grid model, it is not expected to reproduce the exact values of the empirical observation. Quantitative discrepancies are a result of factors that were not incorporated into the model to avoid over-complication, such as: nest-ant behavior (spatial distribution, movement and secondary trophallaxis between nest-ants), the duration of trophallaxis events, and the number of foragers. This point is brought to the readers’ attention in lines 170-174 and 262-264.

Our manuscript presents data on three levels (figure 4-5) empirical observations, a semi-realistic 2D agent based model that captures these observations, and a much-simplified 1D model that agrees with the 2D model and is analytically tractable in a way that provides the desired intuition. We thus felt that there is no need to include a second 2D model that is grid-based and stands midway between the 2D continuous model and the simplified 1D model. To keep the manuscript easy to read, we decided to omit the discrete model included in our original submission from the manuscript and replace it completely with the 2D continuous model. Of course, this denecessitates a comparison between the two 2D models.

If the referees feel that including a grid-based model would still help us make our point we agree to describe it in the SI and make the desired comparisons with the continuous model. Again, we think that the manuscript is clearer without this.

(6) There are a few questionable parameters that the authors have chosen in their model, likely for analytical tractability. For example, the authors assume that at each interaction between a forager and a nest ant, the forager offloads enough food to fill 15% of the crop space remaining in the receiving ant. One can assume that this parameter is something like the 63.21% associated with an exponential time constant or may be based on empirical measurements of transfer in real ants, but the actual justification is not completely clear from the manuscript. Because the mathematical models make predictions that depend on these parameters, their existence (and plausible values) is itself an important assumption that needs to be defended for the argument to be compelling.

The amount of food passed in an interaction is in fact an empirical observation, which was the focus of our previous publication. We observed an exponential distribution of interaction volumes that was scaled to the receiver’s empty crop space, with an average of ~15%. This is mentioned in lines 148-151, and 465-466.

Additionally, the justification for the values of all of the constants used in our mathematical analysis is now mentioned in the mathematical analysis section in the SI (lines 736-742).

Lastly, our new continuous model now contains additional parameters: the velocity of the ants (extracted from the empirical data) and the distance between ants required for trophallaxis to take place (chosen based on the length of the ants’ antennae). This is mentioned in lines 139 and 147.

(7) The behavioral model described by the authors assumes that ants are able to choose a direction toward their nest's entrance at any time. This within-nest path-integration ability does not seem cognitively inexpensive, which narrows the cognitive distance between the behavioral model they propose here and the one they had proposed in their prior work, and weakens the argument for the relevance of this new model. The authors failed to place their work within the context of other simple cue-based motion-switching behaviors discussed in the literature for other taxa -- such as "running" and "tumbling" in *E. coli* bacteria -- but if they had, they might have envisioned an alternative crop-based motion rule that would have the same effect as their current rule (i.e., movement toward the entrance on low crop state) without having to assert that the ant moves directly back towards the entrance. Bridging the work to these other studies would be important here.

Indeed, the new model demands that the ants’ bias towards the nest entrance change with the amount of food in their crop and this requires that the ants be aware of the direction the entrance. This indeed entails some cognitive load on the ants. However, the same cognitive load is also present in the previous models in which, after an ant makes a decision to exit, she quickly (i.e. after a single or, at most, few decision time steps in one version of the model) acts upon it. Acting upon it means knowing the way to the nest entrance. Therefore, the navigational cognitive load is present in all versions of our model and does not narrow the “cognitive distance” between current and previous models. One may still argue that in previous models the forager is relieved of tracking the location of the door before a decision is reached. First, losing oneself and then refinding yourself is not a simple task (see paper by Wehner). Second, the direction to the nest’s entrance may be inferred via gradients of chemicals on the nest surfaces (see reference 50 - Heyman at al, 2017) which may be relatively cheap in the cognitive sense. All these points are now better explained in paragraph 4 of the discussion (lines 306-310).

(8) Focusing on the explanatory power of this model specifically for (some) ants, the authors do not address how to empirically reconcile the ambiguity between the more cognitive mechanisms proposed in their previous work (where ants "decide" to exit a nest) and the current proposal (where the nest cavity "decides" when the ant will exit). For this new hypothesis to be useful, it must be empirically discriminable from the previous hypothesis. At first glance, it is difficult to imagine an experiment that would lead to different predicted behavior from the two different hypotheses. In other words, at the moment, it seems impossible to tell whether the "ant decide" or the "nest decide" model is a better predictor of real ant behavior/cognitive architectures. The lack of discriminability becomes even more problematic when considering that the current version of the model actually increases some cognitive demands by assuming (as described above) that ants keep track of the position of the entrance over the trajectory within the nest.

First, as explained in the answer to the previous remark (7), it is not the case that the new model “actually increases some cognitive demands” in both models the ants have to know the direction to the nest (see paragraph 4 of the Discussion). Furthermore, the new model explains more of the data than the old models: it is spatially explicit and it describes the forager’s motion within the nest both aspects that the old models do not explore (see paragraph 5 of the Discussion). Finally, the old models include a decision to exit at a moment in which exiting is not an available option and this leads to a fuzziness which is liable to lead to inconsistencies. The new model lacks such inconsistencies (see the new paragraph 8 of the discussion, lines 342-353). Therefore, the new model is better defined, it has larger explanatory powers, and it has lower demands on ant cognition. This makes this model a better candidate for explaining the data.

That being said, being a better candidate is not enough to make the new model true. In paragraph 9 of the discussion we now suggest a behavioral experiment which could be used to distinguish between the two mechanisms.

Moreover, we feel that presenting the new model has further merit as it brings forward the idea of single ant stigmergy and demonstrates how it provides a very simple example of exporting cognitive burdens onto physical space (see new paragraph 10 in the discussion, lines 367-383).

(9) In the stochastic model in the Appendix (an integral is used when instead of a sum, perhaps?), it seems like the average values s(Bin) and s(Bout) should depend on F. However, they are treated as constants in Equations (S3) - (S5). If the authors tried to empirically validate the stochastic model by generating many simulated replications and then plotting averages against this prediction, they would likely have a hard time calculating s(Bin) and s(Bout) to generate their numerical predictions. The authors should clarify this point.

An integral is used because the average number of steps on each ant (s(Bin)) may not be an integer. We now mention this explicitly in line 721.

Neglecting boundary effects, the values of s(Bin) and s(Bout) do not depend on F, as mentioned briefly in line 250-252 and more elaborately in the stochastic model section in the SI (lines 702-705 and 743-749). We know this using an absorbing Markov chain analysis, which we now present explicitly at the end of the mathematical explanation. By plugging in the experimental values for all of the constants, we now verify that the equation matches the observed output of the 1D simulation, such that the simplification is sufficient.

(10) "It has recently been suggested that physical space can be utilised to offload computation from individuals' cognition to their environment in the context of collective quorum sensing ([5])." It seems surprising to say that this is the first time this has been suggested. For example, the literature on the effects of nest architecture cited by Reviewer #3 is based on this idea (see below).

Indeed, it is clear that stimergy (e.g. the mentioned literature on nest architecture), an idea that has been around for a long time, can be viewed as colony memory that is etched into the environment. The claim we wish to make in this paper is that the agent has an embodiment and is an actual part of the physical environment. Therefore, one of the simplest ways in which it can alter the environment is by changing its position within it. Comparing this paper to our previous work we show how the ant could relieve itself from computing trophallaxis rates simply by altering its location relative to the nest entrance. It is this idea that is reminiscent of the ideas expressed in [5] wherein computation that is generally assumed to occur in the animal’s brain is externalized to physical space.

More generally, the idea that movement patterns determine encounter rates and thus communication was suggested for ants decades ago. This study seems to sidestep many spatially explicit models on information exchange through encounters (e.g., see review in Gordon 2020 Ann Ent Soc 2020doi: 10.1093/aesa/saaa03).

We did not make any claim about movement patterns determining encounter rates. In fact in our simplified model, they do not and interaction rates are constant throughout the forager’s motion within the nest. Furthermore, in most encounter rate models the ants have to internally compute rates (in one way or another) and then use this measure to alter their behavior. In our model (as the point emphasized in reference [5]), there is no computation or memory going on in the animal’s brain.

We now refine our wording and quote some of the relevant papers in lines 23-29, and contrast them to our simplified model in lines 367-383.

The models use assumptions that ignore the effects of space. The impact of these assumptions should at least be considered.

We did not understand this question or which models it refers to. This is since both the model in our current paper and most models in Gordon et al. (which the previous sentence by the reviewer cites as “spatially explicit”) do take into account space, rather than ignore it.

To summarize, we now remove the claim that reference [5] was the first in this sort of claim. Instead we changed lines 23-32 to be clear about this issue, and add a clearer discussion of the relevant stigmergy literature (see paragraph before last of the Discussion) and, in the same place, make the point that the example that our current work provides is , perhaps, the simplest form of stigmergy – altering a single agent’s own location within space.

(11) The manuscript says nothing about the empirical data which were obtained in another study. This manuscript should say how "average crop load" was measured, including some measure of variation. The manuscript should also say how "foraging frequency" was measured and under what conditions. How often are these conditions likely to occur for colonies of this species?

The experiments are described briefly in lines 84-86 and then referenced to the study that presented them. We now add some details to emphasize that these were conducted in laboratory conditions.

We mention that crop loads were measured using fluorescence imaging (line 85). The wording in the explanation of “average crop state” in the caption of Figure 5 is now changed to be clearer.

The calculation of “foraging frequency” is mentioned in the caption of Figure 4, and now also reiterated in lines 165-166.

Comparison between natural conditions and the experimental conditions that may affect the measured foraging frequency (mainly nest architecture and level of starvation) are discussed in lines 332-340.

(12) What is the "linearity of foraging frequency"? Line 49: "Progress has also been made toward revealing the local mechanism underlying the linearity of foraging frequency, though to a lesser extent. " Again, on lines 105-106: "emergent linear relationship between foraging frequency and total colony hunger." Better definition of this term is important.

We now refrain from saying linearity of foraging frequency and instead refer more directly to the fact/observation that foraging frequency scales linearly with total colony hunger. We made changes in 4-5 places (e.g. the places noted by the referees, the title of the caption to figure 4, and the title of section 3.4) in the manuscript to use this more direct and clear wording.

(13) Figure 5D – empirical results: Why does the forager have a higher crop load at the end of its time inside the nest than at the beginning?

Figure 5D only shows the amount of food in the forager’s crop when she exits the nest, and not when she entered. The vertical axis is the forager’s crop load at the end of each unloading bout, and the horizontal axis is the colony state. It shows that the forager exits the nest with a wide variety of crop loads, the average of which is quite constant across colony states, with a slight rise at high colony states. We changed the wording in line 259-260 to be clearer.

(14) If "hunger" is defined as below as the amount of food in the colony, then it is circular (not "intriguing") to say that the rate at which food comes in matches the total level of "hunger" or level of food.

It is intriguing that *the rate* at which food enters the nest matches the total *amount* of food in the nest. Food could have entered the nest at a constant rate, or at an increasing or decreasing rate that does not correlate linearly with the colony state. The fact that it is proportional to the level of “hunger” is non-trivial and implies the existence of cross-scale feedback between the total amount of food in the colony (on the colony-scale) and the rate of incoming food (on the scale of individual foragers). The emergence of this cross-scale feedback is very intriguing. We now stress this notion in lines 36-37.

(15) It seems strange to cite Oster and Wilson to say that ants collecting food are called foragers. There is at least 100 years of work on ants before Oster and Wilson that referred to "foragers".

We did not mean to say that Oster and Wilson coined the term but rather that one could learn about foragers in this classic reference. We have now cite a book which uses this term 98 years before the book by Oster and Wilson.

(16) Lines 36-38: "Trophallaxis is the main food-sharing method in many ant species 36 ([12]). Each time a laden forager returns to the nest, she unloads the food from her crop to several receivers via 37 trophallaxis. The food further circulates through a complex trophallactic network among all colony members38 ([9], [13]-[17])." This is a misleading way of framing this study, because it equates the distribution of food sources among ants within the nest with the unloading of nectar by foragers. There are many species that use trophallaxis but not directly from foragers.

We changed the wording of this sentence to make it clear that this paper mostly involved forager to non-forager trophallaxis interactions. (lines 41-44)

(17) The results suggest that unloading is associated with whether a forager moves toward or away from the nest entrance. This is called the "deeper nest", but it seems the previous empirical study was performed in a flat arena, and the simulations do not include anything about depth. Thus it gives the impression that an ant associates unloading with going up or down, but in this study, unloading was associated with toward or away from the entrance from an arena. It would be better not to use "deeper" to mean "away from the entrance" as this evokes an image of depth in an ant nest, which is misleading. Since "deep" has an ambiguous meaning here, it is difficult for the reader to know what "deepening" and "lengthening" mean in line 140: "The simulation qualitatively reproduced the lengthening and deepening of foragers' trips."

To avoid this kind of confusion we added a sentence to clarify what we mean by “deep” (line 108), and also refrain from using this word wherever possible.

(18) It was unclear what was meant by: "Note that contrary to the assumptions used in our previous paper ([1]), here a forager never decides to exit the nest. Rather, an exit occurs if the forager's motion brings her to the nest exit." What is the difference between (1) a decision to go to the nest exit and leave the nest, and (2) going to the nest exit and leaving? In the literature on behaviour, decision-making, and cognition, (1) and (2) are the same.

We thank the referees for pointing out the unclarity in the distinction between the two alternatives.

In the context of animal behavior, making a decision means “choosing a specific behavior from a suite of possible ones”. Thus, pinpointing the moment at which an animal takes its decision involves an enumeration of the possible behaviors at that moment. Behaviors involve actions and this means we should look at the various affordances that the environment offers the individual at the time of decision and check which of these are chosen. In option 1 the possible behaviors are either exiting or continuing to interact. In option 2 the possible actions are walking towards or away from the nest entrance. We claim that for an ant that is deep within the nest the option of passing through the entrance, is not within the “suite of possible behaviors”. Therefore, according to the classic definition for decision making as provided above, deciding to exit while deep in the nest (i.e. option 1) carries little meaning.

One could assume, as we implicitly did in our previous work, that once an ant decides to exit she quickly gets to the entrance while avoiding any further interactions and leaves. However, even this assumption cannot be correct. Ants interact in pairs. What would happen if our focal ant has decided to avoid further interactions but, on the way to the entrance, she encounters an ant that has made a decision to interact specifically with her – and initiates such an interaction.

These points make it clear that defining a decision to carry out a behavior that is not currently possible (or pinpointing the moment at which this decision is taken) quickly becomes fuzzy. At some level of detail, will cease being a useful tool in understanding the animal’s behavior. We therefore suggest sticking with the classical definition of a decision in animal behavior and defining the decision to exit as something that occurs when an exit is possible. In our current work, an ant does make a decision far inside the nest. This decision is based on its internal physiological condition. Specifically, based on a threshold in her crop load the ant decides to change her motion characteristics – a decision which is clearly available to her at that point in time.

We now better explain these points in the new paragraph 8 of the discussion.